# Prevalence of pathogenic trypanosome species in naturally infected cattle of three sleeping sickness foci of the south of Chad

Joël Vourchakbé[1,2], Arnol Auvaker Zebaze Tiofack[2], Sartrien Tagueu Kante[2], Padja Abdoul Barka[2], Gustave Simo ⬚[2]*

1 Department of Biological Science, Faculty of Science and Technology, University of Doba, Doba, Chad,
2 Molecular Parasitology and Entomology Unit, Department of Biochemistry, Faculty of Science, University of Dschang, Dschang, Cameroon

* gsimoca@yahoo.fr, gustave.simo@univ-dschang.org

**Data Availability Statement:** All relevant data are within the paper.

**Funding:** This study was funded through the fellowship offered by the "Organisation de

## Abstract

Although a diversity of trypanosome species have been detected in various animal taxa from human African trypanosomosis (HAT) foci, cattle trypanosomosis has not been addressed in HAT foci of west and central African countries including Chad. This study aimed to determine the prevalence of pathogenic trypanosome species in cattle from three HAT foci of the south of Chad. Blood samples were collected from 1466 randomly selected cattle from HAT foci of Mandoul, Maro, and Moïssala in the south of Chad. For each animal, the sex, age and body condition were recorded. Rapid diagnostic test (RDT) was used to search *Trypanosoma brucei gambiense* antibodies while the capillary tube centrifugation (CTC) test and PCR-based methods enabled to detect and identify trypanosome species. From the 1466 cattle, 45 (3.1%) were positive to RDT. The prevalence of trypanosome infections revealed by CTC and PCR-based method were respectively 2.7% and 11.1%. Trypanosomes of the subgenus *Trypanozoon* were dominant (6.5%) followed by *T. congolense* savannah (2.9%), *T. congolense* forest (2.5%) and *T. vivax* (0.8%). No animal was found with DNA of human infective trypanosome (*T. b. gambiense)*. The overall prevalence of trypanosome infections was significantly higher in animal from the Maro HAT focus (13.8%) than those from Mandoul (11.1%) and Moïssala HAT foci (8.0%). This prevalence was also significantly higher in animal having poor body condition (77.5%) than those with medium (11.2%) and good (0.5%) body condition. The overall prevalence of single and mixed infections were respectively 9.4% and 1.6%. This study revealed natural infections of several pathogenic trypanosome species in cattle from different HAT foci of Chad. It showed similar transmission patterns of these trypanosome species and highlighted the need of developing control strategies for animal African trypanosomosis (AAT) with the overarching goal of improving animal health and the economy of smallholder farmers.

Coordination pour la lutte contre les Endémies en Afrique Centrale (OCEAC)", based on the financial cooperation between the CEMAC and the German Federal Ministry for Economic Cooperation and Development (BMZ) and administered by the "Kreditanstalt für Wiederaufbau (KfW)". Is was also funded by the "Alexander von Humboldt Foundation" of Germany through the "Digital Fellowship" offered to Gustave Simo.

**Competing interests:** The authors have declared that no competing interests exist.

## Introduction

African trypanosomoses are parasitic diseases that still have a public health importance. In sub-Saharan Africa, they cause animal African trypanosomosis (AAT) or "nagana" and human African trypanosomosis (HAT) or sleeping sickness respectively in animals and humans. In west and central Africa, *Trypanosoma brucei gambiense* is responsible of the chronic form of HAT that represents about 98% of new reported cases [1]. The remaining 2% of HAT cases is due to *Trypanosoma brucei rhodesiense* which is responsible for the acute form of HAT that occurs in Eastern and southern Africa. Control efforts deployed in the last three decades have significantly reduced the prevalence of HAT in many affected countries and currently, HAT has been included in the World Health Organization (WHO) road map of neglected tropical diseases [1]. With less than 1,000 new HAT cases reported per year, the current target is the interruption of HAT transmission by 2030 [2].

AAT is still widely spread in more than 10 million km2 of land and about 55 millions of cattle, 30 millions of sheep and 40 millions of goats are exposed to the risk of this parasitic disease [3]. It is a chronic debilitating disease affecting livestock and the economy of people living in endemic countries. Yearly, AAT can induce the loss of 10 to 50% of cattle, 2 to 10% of agricultural production, 5 to 30% of meat and 10 to 40% of milk production [3]. The direct and indirect losses linked to AAT can be estimated yearly at 4.5 billion US dollars [4]. In sub Saharan Africa, the pathogenic animal trypanosome species include *Trypanosoma congolense*, *Trypanosoma vivax*, *Trypanosoma simiae* and *Trypanosoma brucei brucei* [5]. Although most of these trypanosome species are transmitted by the tsetse fly which is their biological vector, *Tabanids* and *Stomoxys* can mechanically transmit *T. vivax*, *T. congolense* and *T. brucei* s.l. [6, 7]. AAT is able to constrain the development of sub-Saharan countries by constraining livestock production and posing a threat to household food security and livelihoods. Causing morbidity, mortality, milk and weight loss and inducing significant control and treatment costs, AAT costs a lot of money to the livestock production sector [8].

For number of inhabitants in sub-Saharan African countries, livestock contribute a huge percentage to the Gross Domestic Product and constitute a major source of foreign currency earning [9]. However, trypanosome infections affect about 3 million cattle every year [10] and constitute a threat for the achievement of Sustainable Development Goals like "ending poverty", "zero hunger", "good health and well-being". The fight against trypanosome infections will surely improve animal health, animal and agriculture production as well as the income/savings of smallholder farmers.

Although several investigations have been undertaken on AAT in areas subjected to intensive livestock breeding [11], little attention has been paid to AAT in HAT foci of most central African countries. Most studies on trypanosome infections in these HAT foci aimed to understand animal reservoir by searching *T. b. gambiense* and other trypanosome species in tsetse flies and various animal taxa including small ruminants, pigs, dogs and wild animals [12–15]. Up till now, no attention has been paid to cattle trypanosomiasis in most HAT foci of west and central African countries; probably because cattle breeding was not important for inhabitants of these HAT foci. However, in the Chadian HAT foci that are located in the southern part of the country, environmental conditions are favorable for cattle breeding and tsetse development. In these HAT foci, entomological studies reported several tsetse species including *Glossina tachinoides*, *G. fuscipes fuscipes* and *G. morsitans submorsitans* [16]. In addition, cattle are important for inhabitants of the Chadian HAT foci because they are of great economic value and are also commonly used for traction and transportation. Paying attention for trypanosome infections in cattle of these HAT foci appears important for animal health, animal and agriculture production for smallholder farmers' income. This is strengthened by the fact that several

trypanosome species have been recently reported in horses and donkeys from the Chadian HAT foci [17].

In the progressive control pathway that was recently defined for AAT, it is advocated to enhance research aiming to understand the risks of trypanosomosis. A better understanding of these risks appears not only as the first step of the process that will lead to effective control of AAT, but also as an important component to guide the selection of priority intervention areas [18]. It is in this framework that understanding the epidemiological situation of AAT is becoming very interesting because the lack of accurate information on the prevalence of trypanosome infections constitutes a major drawback to effectively control AAT [19]. Identifying trypanosome species and determining their prevalence in animals from different settings are essential for the understanding of AAT epidemiology and the development of control strategies aiming to eliminate these parasitic diseases.

This study was designed to determine the prevalence of pathogenic trypanosome species in cattle from three HAT foci of the south of Chad.

## Material and methods

### Study site

This study was conducted in the Maro, Mandoul and Moissala HAT foci of the south of Chad [17].

- The HAT focus of Maro (8˚28′33″N, 18˚46′10″E) is situated at 55 km from Sarh, the capital of the "Moyen Chari" Region and at the border of the Central African Republic. In this HAT focus, the temperatures vary from 25 to 38˚ C and the annual rainfall between 800 and 1300 mm. The vegetation is dominated by savannah and clear forests with dotted trees. Inhabitants of this focus practice subsistence agriculture (millet and cassava), fishing, gathering, hunting and breeding of cattle, small ruminants, pigs and equines [20].

- The Mandoul HAT focus (8˚6′57″N, 17˚06′58″E) is located at the borders of Cameroon and the Central African Republic, at about 50 km from Doba, the capital of the "Logone Oriental" Region. In this HAT focus, the temperatures vary between 22 and 38˚ C and the average annual rainfall is 1000 mm [20]. The vegetation is mainly made up of forest galleries and wooded savannah. Inhabitants of this HAT focus practice subsistence farming (culture of cotton, millet and sesame) around the forest galleries and extensive animal breeding of cattle, sheep, goats, pigs and equines. This focus has been subjected to vector control campaign through the deployment of "tiny targets" during three consecutive years (2014 to 2016) [16].

- The HAT focus of Moissala (8˚20′25″N, 17˚45′58″E) is located in the South of Koumra, the capital of the Mandoul Region. Its temperatures vary from 24 to 38˚ C with an average annual rainfall of about 1100 mm. Its vegetation is mainly dominated by forest galleries. Inhabitants of this locality practice subsistence agriculture (culture of cotton, millet and sesame) and breeding of cattle, small ruminants, pigs and equines.

### Sample size estimation

For this cross-sectional study, a stratified sampling strategy was used to select herds and individual animal per herd. Only herds with a minimum of 10 cattle were included. Cattle were sampled by herd and in each herd, blood samples were collected in about 20% of animals. However, more than 20% of animals of some herds were sampled due to the interests and cooperation of some herders and advice from veterinarians. From each chosen herd, the selection of cattle to sample was performed as described by Asgedom et al. [21] using a systematic random sampling technique. The sample size was estimated as described by Thursfield [22].

## Sample collection

Cattle were sampled during three field cross-sectional surveys. The first one was carried out from 7 to 27 March 2019 in the Maro HAT focus; the second from 2 April to 12 May 2019 in the Mandoul HAT focus and the last from 25 May to 14 June 2020 in the Moissala HAT focus. Before the surveys, the objective of the study was explained to inhabitants and local authorities. In each village, only cattle that have been exposed to tsetse fly bites by spending at least three months in the study zone were included and randomly selected for sampling. From each selected cattle, about 5 ml of blood were collected from the jugular vein into EDTA coated tubes. Each tube was labelled and carefully packed. The cattle were the Boro breed [23].

Cattle age was estimated using the dentition pattern as described by Poivey *et al.* [24]. The body condition score (poor, medium and good) of each cattle was determined as described by Nicholson and Butterworth [25].

## Immunologic test

The immunologic test or rapid diagnostic test (RDT) for the gambiense-HAT was performed to see if cattle is infected with *T. b. gambiense*. For this study, the SD BIOLINE HAT test was used. This test was developed using native variable surface glycoproteins (VSGs) (Nat-LiTat 1.3 and Nat-LiTat 1.5) from the Institute of Tropical Medicine (ITM) of Belgium [26]. It quantitatively detects anti-VSG LiTat 1.3 and anti-VSG LiTat 1.5 antibodies of all isotypes (IgG, IgA and IgM) [27, 28]. RDT was performed as described by Matovu et al. [26].

## Parasitological tests

From each blood sample, trypanosome infections were investigated using the capillary tube centrifugation test (CTC) as described by Woo [29]. All animals carrying trypanosome infections revealed by CTC were treated following local veterinarian's advice. Infected cattle were treated with 0.5 mg of Trypamidium per kilogram of body weight supplemented with 0.5 mg of Quinapyramine per kilogram of body weight.

The remaining blood sample was centrifuged at 13,000 rpm for 5 min and the buffy coat was collected from all samples and then, transferred into labelled microtubes. In the field, these tubes were kept in an electric cooler and in the laboratory, they were stored at -20˚C. These tubes were subsequently transported into an electric cooler to the Molecular Parasitology and Entomology Unit of the Faculty of Science of the University of Dschang in Cameroon where they were immediately stored at -20˚C until DNA extraction.

## DNA extraction

Genomic DNA was extracted from each buffy coat sample using the cethyl trimethyl ammonium bromide (CTAB) method as described by Vourchakbe et al. [17].

## Identification of different trypanosome species

This identification was performed as described by Ravel et al. [30]. DNA fragments of internal transcribed spacer 1 (ITS1) of ribosomal DNA of different trypanosome species were amplified in two PCR rounds; the first round was carried out in a final volume of 25 μl containing 1 X PCR buffer, 2 mM $MgCl_2$, 1 μl (10 pmol) of each primer (TRYP18.2C: 5′–GCAAATTGCC–CAATGTCG–3′; TRYP4R: 5′–GCTGCGTTCTTCAACGAA–3′) 0.5 μl (200 mM) of dNTPs, 1 μl (one unit) of *Taq* DNA polymerase (5 U/μl; New England Biolabs), 5 μl of DNA and 14 μl of nuclease free water. For this PCR round, the amplification program consisted of a denaturation step at 94˚ C for 3 min and 30 s. This was followed by 30 amplification cycles and each of

these cycles was made up of a denaturation step at 94˚ C for 30 s, an annealing step at 58˚ C for 1 min, and an extension step at 72˚ C for one minute. A final extension was performed at 72˚ C for 5 min. The amplified products of the first PCR were diluted 10-fold and 3 μl of each dilution was used for the nested PCR. The second PCR was performed using two other primers (IRFCC: 5′–CCTGCAGCTGGATCAT–3′ and TRYP5RCG: 5′–ATCGCGACACGTTGTG–3′). The PCR conditions and the amplification program were identical to those of the first PCR.

After the nested PCR, PCR products were separated by electrophoresis on a 2% agarose gel that was subsequently stained with ethidium bromide (0.3 μg/ml) and visualized under UV light. Trypanosome species were identified on the basis of length polymorphism of ITS1 fragments. DNA fragments of around 630 bp (630 bp for *T. congolense* forest and 610 bp for *T. congolense* savannah) were expected for *T. congolense* while fragments of about 150 bp and 400 bp were expected respectively for *T. vivax* and trypanosomes of the sub-genus *Trypanozoon* (*T. brucei* s.l. or *T. evansi* or *T. equiperdum*).

## Differentiation between *Trypanosoma congolense* forest and *Trypanosoma congolense* savannah

After amplification of ITS1 fragments of different trypanosome species, all samples that revealed DNA fragments between 600–650 bp corresponding to the expected size for *T. congolense* were selected and then, subjected to other PCRs that aimed to differentiate *T. congolense* forest from *T. congolense* savannah. For this differentiation, specific primers for *T. congolense* forest or *T. congolense* savannah were used to identify each of these trypanosomes. The differentiation was performed as described by Simo *et al.* [13] using TCF1 (5′–GGA CACACGC CAGAAGGTACTT–3′) and TCF2 (5′–GTTCTCTCGCACCAAATCCAAC–3′) primers for *T. congolense* forest [31], and TCS1 (5′–CGAGCGAGAACGGGCAC–3′) and TCS2 (5′–GGGA CAAACAAATCCCGC–3′) primers for *T. congolense* savannah [32]. PCR reactions were performed in a final volume of 25 μl containing 1 X PCR buffer, 3 mM MgCl₂, 1 μl (15 pmol) of each primer, 0.5 μl (200 mM) of dNTPs, 1 μl (one unit) of *Taq* DNA polymerase, 3 μl of DNA and 16 μl of sterile water. The amplification program comprised a denaturation step at 94˚ C for 3 min 30 s, followed by 40 amplification cycles. Each of these cycles included a denaturation step at 94˚ C for 30 s, an annealing step at 60˚ C for one minute and an elongation step at 72˚ C for one minute. This was followed by a final elongation step at 72˚ C for 5 minutes.

Amplicons were resolved by electrophoresis on 2% agarose gel that was stained with ethidium bromide (0.3 μg/ml). DNA fragments were visualized under ultraviolet light and then photographed.

## Search for *Trypanosoma brucei gambiense* infections

This was performed on all samples that have shown a DNA fragment of about 400 bp, corresponding to the expected size of trypanosomes of the subgenus *Trypanozoon* (*T. b. brucei*, *T. evansi*, *T. b. gambiense* and *T. b. rhodesiense*). *Trypanosoma b. gambiense* infections was investigated using a nested PCR as described by Cordon-Obras *et al.* [33]. During this investigations, two pairs of *T. b. gambiense* specific primers were used: TgSGP1 (5′–GCT GCT GTG TTC GGA GAG C–3′ and TgSGP2- (5′–GCC ATC GTG CTT GCC GCT C–3′) described by Radwanska et al. [34], and TgsGPs (5′–TCA GAC AGG GCT GTA ATA GCA AGC–3′) and TgsGPas (5′–GGG CTC CTG CCT CAA TTG CTG CA–3′) designed by Morrison *et al.* [35]. The first PCR was carried out as described by Vourchakbe *et al.* [17]. It was performed in a total volume of 25 μl containing 2.5 μl of 10 X PCR buffer (10 mM Tris-HCl (pH 9.0), 50 mM KCl, 3 mM MgCl₂), 1 μl (15 pmol) of each primer (TgSGP1 and TgSGP2), 0.5 μl (100mM) of dNTPs, 1 μl (one unit) of Taq DNA polymerase, 5 μl of DNA and 14 μl of sterile

water. The amplification program was made up of a denaturation step at 95˚C for 3 min followed by 45 cycles of 95˚C for 30 s, 63˚C for 1 min and 72˚C for 1 min. A final elongation step was performed at 72˚C for 5 min. Amplicons of the first PCR were diluted 10 fold and 5 μl of each dilution was used as DNA template for the nested PCR. For this PCR, TgsGPs and TgsGPas primers were used and only 25 amplification cycles were performed in the same conditions as for the first PCR.

Amplicons of the nested PCR were resolved by electrophoresis on a 2% agarose gel stained with ethidium bromide (0.3 μg/ml). After electrophoresis, DNA fragments were visualized under UV light and then photographed.

### Ethical considerations

The protocol of this study was approved by the Bioethics Committee of the Ministry of High Education, Research and Innovation of Chad with the reference number 462/PR/PM/MESRI/SG/CNBT/2017. The review board of the Molecular Parasitology and Entomology Subunit of the Faculty of Science of the University of Dschang gave also its approval. The local administrative, religious and traditional authorities of different HAT foci also approved this study after detailed explanation of its objectives. Verbal consent was obtained from animal owners after explaining to each of them the aim of the study.

### Data analysis

The k-proportion test or chi-square test of equality of proportion was used to compare the prevalence of different trypanosome species and also the prevalence of trypanosome infections between HAT foci. Each comparison was considered significant when the P value was <0.05. Multiple logistic regression models were used to estimate odd ratio (OR) and 95% confidence intervals (CI) for the association between HAT foci, body condition, age groups, sex and trypanosome infections.

## Results

### Characteristics of the study population

For this study, 1466 cattle including 876 (59.8%) females and 590 (40.2%) males were sampled in three HAT foci: 468 (31.9%) from the Mandoul HAT focus, 513 (35.0%) from Maro and 485 (33.1%) from Moïssala (Table 1). Of these 1466 cattle, 1127 (76.9%) were aged two years or over while 339 (23.1%) were calves (cattle of two or less than two years old) (Table 2). One hundred and seventy three (11.8%) of these cattle had poor body condition while 205 (14.0%) and 1088 (74.2%) had medium and good body condition, respectively (Table 2).

### Results of serological tests

RDT was positive in 45 cattle (3.1%): 10 (2.1%) cattle from the Mandoul HAT focus, 22 (4.3%) from Maro and 13 (2.7%) from Moïssala (Table 1). The positivity rate of RDT varied between HAT foci (Table 1). However, comparing these positivity rates of RDT, no significant difference ($\chi^2$ = 0.066; p = 0.967; 95%CI: 2.2–4.0) was observed between HAT foci (Table 1).

The percentage of positive RDTs vary according to age groups and body condition. It was 2.1% in calves and 3.4% in older cattle. This percentage was significantly higher ($\chi^2$ = 188.9; p < 0.0001; 95%CI: 2.2–4.0) in cattle having poor body condition (19.7%) than those with medium (3.9%) and good body (0.3%) condition (Table 2). It was 3.1% in males as well as in females, 2.06% in calves and 3.37% in older cattle (Table 2). Comparing the percentages of cattle that were positive to RDT, no significant difference was recorded between male and female

**Table 1. Prevalence of trypanosome species according to HAT foci.**

| HAT focus | NE | Serology | | Parasitology | | PCR results | | | | | | | | | | | | | |
|---|---|---|---|---|---|---|---|---|---|---|---|---|---|---|---|---|---|---|---|
| | | RDT +(%) | 95% CI | T+(%) | 95% CI | TB +(%) | 95% CI | TC+(%) | 95% CI | TCF +(%) | 95% CI | TCS +(%) | 95% CI | TV +(%) | 95% CI | Total (%) | 95% CI |
| Mandoul | | 10 (2.1) | 0.8–3.4 | 9 (1.9) | 0.7–3.1 | 39 (8.3) | 5.8–10.8 | 14 (3.0) | 1.5–4.5 | 9 (1.9) | 0.7–3.1 | 5 (1.1) | 0.2–2 | 7 (1.5) | 0.4–2.6 | 52 (11.1) | 8.3–13.9 |
| Maro | 468 | 22 (4.3) | 2.5–6.1 | 20 (3.9) | 2.2–5.6 | 25 (4.9) | 3.0–6.8 | 53 (10.3) | 7.7–12.9 | 18 (3.5) | 1.9–5.1 | 35 (6.8) | 4.6–9.0 | 4 (0.8) | 0.0–1.6 | 71 (13.8) | 10.8–16.8 |
| Moïssala | 513 | 13 (2.7) | 1.3–4.1 | 10 (2.1) | 0.8–3.4 | 32 (6.6) | 4.4–8.8 | 12 (3.3) | 1.7–4.9 | 9 (1.9) | 0.7–3.1 | 3 (0.6) | 0.1–1.3 | 1 (0.2) | -0.2–0.6 | 39 (8.0) | 5.6–10.4 |
| Total | 485 | 45 (3.1) | 2.2–4.0 | 39 (2.7) | 1.9–3.5 | 96 (6.5) | 5.2–7.8 | 79 (5.4) | 4.2–6.6 | 36 (2.5) | 1.7–3.3 | 43 (2.9) | 2.0–3.8 | 12 (0.8) | 0.3–1.3 | 162 (11.1) | 9.5–12.7 |
| $\chi^2$ | 1466 | 4.17 | 1.3–4.1 | 4.69 | 1.9–3.5 | 4.79 | | 37.9 | | 3.65 | | 42.10 | | 4.9 | | 8.5 | |
| P-value | | 0.12 | 2.2–4.0 | 0.096 | | 0.091 | | < 0.0001 | | 0.16 | | < 0.0001 | | 0.09 | | 0.014 | |

The number of positive PCRs is indicated with their percentages in parentheses; HAT: Human African trypanosomosis; NE: Number of animals examined; RDT: Rapid diagnosis test; T+: Trypanosome infections revealed by capillary tube centrifugation; TB+: trypanosomes of the subgenus *Trypanozoon*; TCF: *Trypanosoma congolense* forest; TC: *Trypanosoma congolense* (including *Trypanosoma congolense* savannah and forest); TCS: *Trypanosoma congolense* savannah; TV: *Trypanosoma vivax*; $\chi^2$: chi-square test.

($\chi^2$ = 0.0; p = 1; 95%CI: 2.2–4.0) as well as between age groups ($\chi^2$ = 1.08; p = 0.29; 95%CI: 2.2–4.0) (Table 2). With ORs of 0.1 (95% CI: 0.02–0.23; p-value = 0.0001) and 0.01 (95% CI: 0.003–0.037; p-value < 0.0001) and significant P values, cattle having good body condition were less likely to be positive for RDT than those with medium and poor body condition, respectively (Table 2).

**Table 2. Positivity of RDTs and prevalence of trypanosome infections according to age groups, sex and body condition of cattle.**

| Variable | NE | RDT+ (%) | 95% CI | P value | $\chi^2$ | OR (%95) | P value* | NIA (%) | 95% CI | P | $\chi^2$ | OR (%95) | P value* |
|---|---|---|---|---|---|---|---|---|---|---|---|---|---|
| **Age groups** | | | | | | | | | | | | | |
| < 2 years | 339 | 7 (2.1) | 0.6–3.6 | 0.29 | 1.08 | - | - | 42 (12.4) | 8.9–15.9 | 0.42 | 0.63 | - | - |
| > 2 years or over | 1127 | 38 (3.4) | 2.3–4.5 | | | 0.6 (0.27–1.36) | 0.22 | 120 (10.6) | 8.8–12.4 | | | 1.19 (0.82–1.72) | 0.37 |
| Total | 1466 | 45 (3.1%) | 2.2–4.0 | | | | | 162 (11.1%) | 9.5–12.7 | | | | |
| **Sex** | | | | | | | | | | | | | |
| Male | 590 | 18 (3.1) | 1.7–4.5 | 1 | 0.0 | - | - | 68 (11.5) | 2.0–4.2 | 0.7 | 0.15 | - | - |
| Female | 876 | 27 (3.1) | 2.0–4.2 | | | 0.99 (0.5–1.81) | 0.97 | 94 (10.7) | 8.7–12.7 | | | 1.08 (0.78–1.5) | 0.43 |
| Total | 1466 | 45 (3.1%) | 2.2–4.0 | | | | | 162 (11.1%) | 9.5–12.7 | | | | |
| **Body condition** | | | | | | | | | | | | | |
| Good | 1088 | 3 (0.3) | 0.0–0.6 | <0.0001 | 188.9 | - | - | 134 (77.5) | 75.0–80.0 | < 0.0001 | 900.3 | - | - |
| Medium | 205 | 8 (3.9) | 1.2–6.6 | | | 0.1 (0.02–0.23) | 0.0001 | 23 (11.2) | 6.9–15.5 | | | 0.04 (0.014–0.09) | < 0.0001 |
| Poor | 173 | 34 (19.7) | 13.8–25.6 | | | 0.01 (0.003–0.037) | < 0.0001 | 5 (0.5) | -0.6–1.6 | | | 0.0013 (0.001–0.003) | < 0.0001 |
| Total | 1466 | 45 (3.1) | 2.2–4.0 | | | | | 162 (11.1) | 9,5–12,7 | | | | |

NE: number of animals examined; NIA: Number of infected animals; OR: odd ratio; P: P value; RDT+: Number of cattle positive to rapid diagnostic test; $\chi^2$: chi square test; CI: confident interval; p-value*: association study; p-value.

## Results of parasitological tests

The CTC test revealed 39 (2.7%) cattle with trypanosome infections (Table 1). Specific identification of different trypanosome species was not possible at this stage. Of the 39 cattle carrying trypanosome infections revealed by CTC, 9 (1.9%) were from the Mandoul HAT foci, 20 (3.9%) from Maro and 10 (2.1%) from Moïssala (Table 2). Comparing the prevalence of trypanosome infections revealed by CTC, no significant difference ($\chi^2$ = 4.69; p = 0.096; 95%CI: 1.9–3.5) was recorded between HAT foci (Table 1).

## Results of molecular tests: Trypanosome species identified

Out of 1466 cattle analyzed in this study, PCR targeting ITS-1 fragments of different trypanosome species revealed trypanosome DNA in 162 of them; giving an overall prevalence of 11.1% (Table 1). A variety of pathogenic trypanosome species including *T. vivax*, *T. congolense* forest and savannah, and trypanosomes of the sub-genus *Trypanozoon* were identified in cattle from the three HAT foci. Trypanosomes of the sub-genus *Trypanozoon* had the highest prevalence of 6.5% while infections of *T. vivax* were least prevalent with a prevalence of 0.8% (Table 1). The overall prevalence of *T. congolense* was 5.4%: 2.5% for *T. congolense* forest and 2.9% for *T. congolense* savannah.

## Prevalence of trypanosome species according to HAT foci

The prevalence of trypanosome species varied according to HAT foci. Cattle from the Maro HAT focus were more infected with trypanosomes than those from other HAT foci (Table 1). When the overall prevalence of trypanosome infections were compared, a significant difference ($\chi^2$ = 8.5; p = 0.014) was recorded between HAT foci (Table 1).

The prevalence of *T. vivax* was low (0.81%) in cattle from all HAT foci: 1.5% in cattle from the Mandoul HAT focus, 0.8% in those from Maro and 0.2% in those from Moïssala (Table 1). Comparing the prevalence of *T. vivax* infections, no significant difference ($\chi^2$ = 4.9; p = 0.09; 95%CI: 0.3–1.3) was recorded between the three HAT foci (Table 1).

The highest prevalence (8.3%) of trypanosomes of the sub-genus *Trypanozoon* was obtained in cattle from the Mandoul HAT focus followed by those from Moïssala (6.6%) and Maro (4.9%) HAT foci. Between these HAT foci, no significant difference ($\chi^2$ = 4.79; p = 0.091; 95% CI: 5.2–7.8) was observed in the prevalence of trypanosomes of the sub-genus *Trypanozoon* (Table 1).

For all *T. congolense*, cattle from the Maro HAT focus (10.3%) were significantly ($\chi^2$ = 37.9; p < 0.0001; 95%CI: 4.2–6.6) more infected than those from the Mandoul (3.0%) and Moïssala (3.3%) HAT foci. Cattle from the Maro HAT focus (3.50%) were more infected with *T. congolense* forest than those from Mandoul (1.9%) and Moïssala (1.9%) HAT foci. Comparing the prevalence of *T. congolense* forest, no significant difference ($\chi2$ = 3.65; p = 0.16; 95%CI: 1.7–3.3) was recorded between HAT foci. This difference was significant ($\chi^2$ = 42.1; p < 0.0001; 95%CI: 2.0–3.8) for *T. congolense* savannah; cattle from the Maro HAT focus (6.8%) being more infected than those from the Mandoul (1.1%) and Moïssala (0.6%) HAT foci (Table 1).

## Prevalence of trypanosome infections according to age groups, sex and body condition

Out of 1127 cattle being more than two years, 120 (10.6%) had trypanosome infections. Amongst the 339 calves (cattle of two or less than two years old), 42 (12.4%) were found with trypanosome infections. Comparing the prevalence of trypanosome infections, no significant difference ($\chi^2$ = 0.63; p = 0.42; 95%CI: 9.5–12.7) was observed between age groups (Table 2).

The prevalence of trypanosome infections were 11.5% in male and 10.7% in female. Comparing these prevalence of trypanosome infections, no significant difference ($\chi^2$ = 0.15; p = 0.7; 95%CI: 9.5–12.7) was recorded between sexes (Table 2).

The 1466 cattle examined in this study were subdivided into three groups: 1088 (74.2%) having good body condition, 205 (4.0%) with medium and 173 (11.8%) with poor body condition. The highest prevalence of trypanosome infections of 77.5% was recorded in cattle having poor body condition and the lowest prevalence of 0.5% in those having good body condition (Table 2). In cattle having medium body condition, the prevalence of trypanosome infections was 11.2% (Table 2). The prevalence of trypanosome infections differs significantly ($\chi^2$ = 900.3; p < 0.0001; 95%CI: 9.5–12.7) according to cattle body condition (Table 2). With ORs of 0.04 (95% CI: 0.014–0.09, p-value < 0.0001) and 0. 0013 (95% CI: 0.0005–0.0035; p-value < 0.0001), cattle in good body condition had a reduced risk to be infected by trypanosomes compared to those with meduim and poor body condition, respectively (Table 2).

## Single and mixed infections of trypanosome species

From 162 cattle carrying trypanosome infections, 138 (85.2%) had single infections while 24 (14.8%) had mixed infections (Table 3). The 138 single infections included 82 (59.4%: 82/138) trypanosomes of the sub-genus *Trypanozoon*, 20 (14.5%: 20/138) *T. congolense* forest, 29 (21.0%: 29/138) *T. congolense* savannah and 7 (5.1%: 7/138) *T. vivax* (Table 3). The overall prevalence of single infections was 9.4% (138/1466) (Table 3). The number of these single infections varied according to HAT focus. Single infections of trypanosomes were predominant in cattle from the Maro HAT focus (37.68%: 52/138) followed by those from the Mandoul (36.23%; 50/138) and the Moïssala (26.08%; 36/138) HAT foci (Table 3).

Mixed infections were made up of one triple infection and 23 double infections. The overall prevalence of mixed infections was 1.6% (24/1466). The triple infection involved trypanosomes of the sub-genus *Trypanozoon*, *T. vivax* and *T. congolense* forest. Trypanosomes of the sub-

**Table 3. Single and mixed infections of trypanosome species according to HAT foci.**

| Type of infections | Trypanosome species | Number of single and mixed infections in each HAT focus | | | | | | TNIC | |
| | | Mandoul | | Maro | | Moïssala | | All HAT foci | |
| | | NI (%) | 95% CI | NI (%) | 95% CI | NI (%) | 95% CI | NI (%) | 95% CI |
|---|---|---|---|---|---|---|---|---|---|
| Single | TB | 32 (39.0) | 27.9–50.2 | 23 (28.0) | 17.7–38.4 | 27 (32.9) | 22.1–43.7 | 82 (50.6) | 42,6–58,6 |
| | TCF | 5 (25.0) | 3.5–46.5 | 12 (60.0) | 36.0–84.0 | 3 (15.0) | 3.1–33.1 | 20 (12.3) | 6,9–17,7 |
| | TCS | 9 (31.0) | 12.5–49.6 | 15 (51.7) | 31.8–71.6 | 5 (17.2) | 1.8–32.7 | 29 (17.9) | 11,7–24,1 |
| | TV | 4 (57.1) | 13.3–100.9 | 2 (28.6) | 12.0–69.2 | 1 (14.3) | 18.8–47.3 | 7 (4.3) | 0,9–7,7 |
| Total | | **50 (36.2)** | **27.8–44.6** | **52 (37.7)** | **29.2–46.1** | **36 (26.1)** | **18.4–33.8** | **138 (85.2)** | **79,4–91,0** |
| Double | TB and TCF | 1 (16.7) | 21.5–54.8 | 4 (66.7) | 20.6–112.7 | 1 (16.7) | 21.5–54.8 | 6 (3.7) | 0,5–6,9 |
| | TB and TCS | 3 (60.0) | 7.1–112.9 | 1 (20.0) | 24.1–65.1 | 1 (20.0) | 24.1–65.1 | 5 (3.1) | 0,1–6,1 |
| | TB and TV | 2 (100) | 75–125 | 0 (00.0) | - | 0 (00.0) | - | 2 (1.2) | -0,8–3,2 |
| | TCF and TCS | 4 (50.0) | 9.1–90.9 | 3 (37.5) | 2.3–77.3 | 1 (12.5) | 16.7–41.7 | 8 (4.9) | 1,3–8,5 |
| | TCF and TV | 0 (00.0) | - | 1 (100) | 50–150 | 0 (00.0) | - | 1 (0.6) | -0,9–2,1 |
| | TCS and TV | 1 (100) | 50–150 | 0 (00.0) | - | 0 (00.0) | - | 1 (0.6) | -0,9–2,1 |
| Total | | **11 (47.8)** | **25.3–70.4** | **9 (39.1)** | **17–61.2** | **3 (13.0)** | **2.9–29** | **23 (14.2)** | **8,5–19,9** |
| Triple | TB, TCF and TV | 0 (00.0) | - | 1 (100) | 50–150 | 0(00.0) | - | 1 (0.6) | -0,9–2,1 |
| Total | | **0 (00.0)** | **-** | **1 (100)** | **50–150** | **0 (00.0)** | **-** | **1 (0.6)** | **-0,9–2,1** |

TB+: trypanosome of the subgenus *Trypanozoon*; TCF: *Trypanosoma congolense* forest, TCS: *Trypanosoma congolense* savannah; TV: *Trypanosoma vivax*; HAT: Human African trypanosomosis; TNIC: total number of infected cattle; NI: number of cattle infected in each HAT focus; %: percentage of cattle infected.

genus *Trypanozoon* were involved in 13 double infections: 6 with *T. congolense* forest, 5 with *T. congolense* savannah and 2 with *T. vivax. Trypanosoma congolense* forest was found in other 9 double infections involving 8 with *T. congolense* savannah and one with *T. vivax*. The last double infection was made up of *T. congolense* savannah and *T. vivax* (Table 3).

### *Trypanosoma brucei gambiense* infections

From the 96 cattle carrying trypanosomes of the sub-genus *Trypanozoon*, the two sets of primers used to identify *T. b. gambiense* infections did not amplify DNA fragment of this trypanosome sub-species: no cattle of the three HAT foci was therefore found with *T. b. gambiense* infections.

## Discussion

Although a variety of trypanosome species have been reported in tsetse fly and various domestic and wild animal taxa from HAT foci of west and central African countries including Chad [12–15, 17, 36, 37], little attention has been paid to cattle and consequently, the problem of cattle trypanosomosis has not been addressed. It is to fill this knowledge gap that the prevalence of different trypanosome species were determined in cattle with the main goal of updating epidemiological data on AAT in Chad. This study on cattle trypanosomosis in central African HAT foci revealed several pathogenic trypanosome species including *T. congolense*, *T. vivax* and trypanosomes of the subgenus *Trypanozoon* in cattle from three Chadian HAT foci. These results highlighting different species of trypanosomes in cattle are in agreement with those reporting such infections in tsetse fly, donkeys and horses of tsetse infested areas of Chad [15, 17, 36, 37].

The RDT used in the present study is an immune-chromatographic test that was designed for the screening of HAT. It is expected to be positive only when the host has been infected by *T. b. gambiense* [38]. Our RDT results suggest that 3.1% of analyzed cattle have probably been infected by *T. b. gambiense*. This is low compared to 18.9% and 19.4% recorded respectively in equines and small ruminants of the same HAT foci [15, 17]. The positivity of RDT could be explained by the fact that cattle can sustain *T. b. gambiense* infections for more than 50 days [38, 39]. During this period, antibodies against trypanosome antigens can be produced by cattle. In addition to that, the possibility for cattle to be infected by *T. b. gambiense* has been demonstrated by cyclical development of *T. b. gambiense* from cattle [39] as well as the isolation from cattle of trypanosomes of the subgenus *Trypanozoon* that were resistant to human serum [38]. Despite the positivity of RDT, no animal was found infected by *T. b. gambiense* since all PCRs targeting this parasite were negative. These results could be explained by the fact that a positive PCR can be interpreted as an active infection although problems of reproductibility of PCR for the diagnostic of HAT have been highlighted by Koffi et al [40]. For cattle that were positive to RDT and negative for the PCR targeting *T. b. gambiense*, the possibility of past infections and self-cure cannot be ruled out because such phenomena have been observed in other animal taxa like pigs [41]. Some positive RDTs could result from cross-reactions with epitopes of other trypanosome species [26]. Indeed, sera from animals infected with either *T. congolense*, *T. b. brucei* or *T. evansi* cross-reacted with antigens used in the RDT and the card agglutination test for *gambiense-HAT* [42–44]. Moreover, it has been demonstrated that sera from both *T. b. gambiense* and *T. b. rhodesiense* patients cross-reacted with a large number of trypanosome antigens including VSG LiTat 1.3 and VSG LiTat 1.5 [45]. Results of the present study confirm the low sensitivity and specificity of RDT for the detection of *T. b. gambiense* infections in cattle as already reported by Matovu et al. [26]. These authors reported that RDTs detect cross-reacting antibodies in cattle.

The overall low prevalence of trypanosome infections of 2.7% revealed by CTC compared to 11.1% recorded with PCR-based method suggests that most infected cattle were carriers of low parasitaemia that was below the detection threshold of the CTC. These results confirm the higher sensitivity and specificity of the PCR-based method for efficient detection of different trypanosome species. It is important to point out that PCR can detect transient infections and a PCR positive result indicates the presence of the corresponding parasite DNA and not necessary an active infection [46]. Despite the higher specificity of the PCR-based method, the differentiation of trypanosomes of the subgenus *Trypanozoon* that include *T. b. rhodesiense*, *T. b. brucei*, *T. b. gambiense*, *T. evansi* and *T. equiperdum* remains a challenge because no single test is able to unequivocally distinguish these trypanosome species [42]. Nevertheless, *T. b. rhodesiense* is not expected because the Chadian HAT foci are located in areas not endemic for the acute form of HAT. The presence of *T. equiperdum* is also unlikely because this parasite is an equine trypanosome transmitted via coitus. Moreover, *T. b. gambiense* was not detected in cattle despite its identification in other animal taxa from the same HAT foci [15, 17]. These results suggest that cattle cannot be considered as potential animal reservoir for the *gambiense*-HAT in the south of Chad. They also suggest that cattle carrying trypanosomes of the sub-genus *Trypanozoon* could be infected either by *T. evansi* or *T. b. brucei* or a combination of these subspecies. The higher prevalence of trypanosomes of the sub-genus *Trypanozoon* indicates high transmission of either *T. b. brucei* or *T. evansi* or a combination of these species in the three HAT foci. Our results highlighting no significant difference in the prevalence of trypanosomes of the subgenus *Trypanozoon* suggests similar transmission patterns in the three Chadian HAT foci.

The detection of trypanosomes of the subgenus *Trypanozoon* is in agreement with results reporting them in most HAT foci in West and Central Africa [12, 13, 47]. In cattle sampled in this study, the prevalence of 6.5% recorded for trypanosomes of the subgenus *Trypanozoon* is lower than 31.6%, 26.7% and 22.4% reported in other animal taxa (donkeys, horses, pigs, dogs and small ruminants) from the same HAT foci [15, 17]. The discrepancies between these results could be explained by the breeding system that varies according to animal taxa and HAT foci, the differences in the susceptibility of animal taxa to trypanosomes of the subgenus *Trypanozoon* and the composition and density of vectors in each HAT focus. Cattle populations can be sedentary and/or transhumant while other animal taxa such as equines, dogs, pigs and small ruminants live permanently in villages of HAT foci. Being regularly exposed to tsetse fly bites, equines and other animal taxa can easily acquire trypanosome infections compared to transhumant cattle that move from tsetse fly infested areas to tsetse fly free areas.

The highest prevalence of trypanosomes of the subgenus *Trypanozoon* in cattle from the Mandoul HAT focus is surprising because the "tiny targets" deployed in this focus for vector control during three consecutive years (2014 to 2016) before our sampling were expected to stop trypanosomes' transmission [16]. This hypothesis was strengthened by results of mechanistic transmission model suggesting that HAT transmission would have been interrupted in 2015 due to intensified interventions in the Mandoul HAT focus [48]. However, from 2016 to 2020, HAT cases have been reported in all Chadian HAT foci including the Mandoul HAT focus where the number of HAT cases decreased while a slight increase was observed in the Maro HAT focus [49]. This continuous detection of HAT cases in the Mandoul HAT focus could be explained by the chronicity of the *gambiense*-HAT since some infected individuals can carry the parasite for many years without being detected. Nevertheless, we cannot rule out the probability of having not only a slight cyclical transmission of trypanosomes to human and mammals by potential residual tsetse fly populations, but also a mechanical transmission of trypanosomes of the subgenus *Trypanozoon* like *T. evansi* to cattle [6, 7]. In addition to that, the movement of herders with some infected cattle in the search of pastures during the

transhumance phenomenon is another factor that could explain the highest prevalence of trypanosomes of the subgenus *Trypanozoon* in cattle from the Mandoul HAT focus.

The overall very low prevalence of *T. vivax* of 0.81% reported in cattle from the three HAT foci corroborates with results obtained in horses and donkeys of the same HAT foci [17]. These results indicate low transmission of *T. vivax* in the south of Chad. In these tsetse infested areas, *T. vivax* is probably of less epidemiological importance than other trypanosome species. Compared to *T. vivax*, the higher prevalence *T. congolense* and trypanosomes of the subgenus *Trypanozoon* might be linked to their effective transmission by *Glossina tachinoides*, *G. fuscipes fuscipes* and *G. morsitans submorsitans* that have been identified in HAT foci of Chad [50].

The present study revealed *T. congolense* forest and savannah in cattle from the three HAT foci of Chad. The co-existence of these trypanosomes (*T. congolense* forest and savannah) in cattle is in agreement with results obtained in equines from the same HAT foci [17]. This co-existence indicates that the geographical limit of these trypanosome species (*T. congolense* savannah and forest in the savannah and forest zones, respectively) tends to change with time. Our results showing a comparable prevalence of *T. congolense* savannah (2.9%) and *T. congolense* forest (2.5%) is difficult to explain because the localization of the three Chadian HAT foci in the forest galleries would have suggested higher prevalence of *T. congolense* forest compared to *T. congolense* savannah. To explain results of the present study, we can speculate about the transhumance phenomenon in which cattle from other tsetse fly infected areas migrate in the southern part of Chad for grazing. During this phenomenon, transhumant cattle that were infected elsewhere with *T. congolense* savannah migrate to HAT foci. These cattle could ensure the transmission of *T. congolense* savannah in other tsetse fly infested areas. This could increase the prevalence of this trypanosome species in tsetse fly and different animal taxa. Nevertheless, the differences in the susceptibility of different animal taxa to *T. congolense* infections cannot be ruled out.

Between HAT foci, the significant differences observed either for the overall prevalence of *T. congolense* or for the prevalence of each *T. congolense* (forest or savannah) indicates some variations in the transmission patterns of these trypanosomes. The high prevalence of *T. congolense* (*T. congolense* forest and savannah) in cattle from the Maro HAT focus and its low prevalence in animals from other HAT foci could be explained by some differences in the control measures that have been implemented in these three HAT foci. The low prevalence of *T. congolense* in cattle from the Mandoul and Moïssala HAT foci results probably from vector control implemented in these foci through the deployment of tiny targets since 2015. As already mentioned above, some differences in the susceptibility of trypanosomes to animal taxa are additional arguments to explain the variations observed in the prevalence of trypanosome infections.

The insignificant difference recorded in the prevalence of trypanosome infections between sex of cattle agrees with results of previous studies reporting equal infections in male and female from other African countries [51]. These results could be explained by the grazing system in which all sex groups move together on grazing land. In such system, female and male are exposed to the same level of tsetse fly bites. However, studies in Ethiopia reported significantly higher prevalence of trypanosome infections in female (29.3%) compared to male (18.5%) cattle [52]. These results indicate that the prevalence of trypanosome infections in male and female cattle can vary according to epidemiological settings.

Our results showing significantly higher prevalence of trypanosome infections in older cattle compared to young ones are consistent with previous data highlighting lower disease prevalence in younger cattle than older ones [52]. These results could be explained by the higher exposure of older cattle to tsetse bites compared to younger ones [53]. In fact, younger cattle are often kept in the farmstead and do not venture far for grazing and watering. This reduces

their exposure to tsetse bites and hence, to trypanosome infections. However, older cattle travel great distances to search for grazing and watering spots. They are more exposed to tsetse fly bites and therefore, to a higher risk of trypanosome infections.

The significantly higher prevalence of trypanosome infections in cattle having poor body condition compared to those with medium and good body condition is consistent with previous data reporting higher trypanosomosis prevalence in cattle with poor body condition than in cattle with medium and good body conditions [52, 53]. These results are strengthened by the low values of odd ratio indicating a decrease risk to be carriers of trypanosome infections for cattle having good body condition compared to those with medium and poor body condition. The higher prevalence can be attributed to the decreased ability of emaciated cattle or cattle having poor body condition to defend against trypanosome infections compared to medium and good-body-condition cattle. Poor body condition can be attributed to trypanosome infections because it is a sign of AAT in livestock. Nevertheless, it is important to point out that animals can become emaciated due to poor nutritional status or other concurrent parasitic infections like helminth infections. In such context, these cattle are more likely to contract trypanosome infections and other diseases because they are not able defend themselves due to their weak immune system. Despite the fact that anaemia is a common feature of AAT, the correlation between the body condition and anemic status of cattle was not assessed because the anemic status of cattle was not recorded in the present study.

Although single infections were predominant (74.80%) in cattle, several of these animals had mixed infections of different trypanosome species. Our results highlighting single and mixed infections of different pathogenic trypanosome species are consistent with data of previous studies reporting such types of infections in tsetse and various animal taxa [12, 13, 54, 55]. The percentage of mixed infections reported here was probably underestimated because some mixed infections could exist between trypanosomes (*T. evansi* and *T. brucei* s.l.) of the sub-genus *Trypanozoon*. The mixed infections identified in these studies could results from sequential transmissions of various trypanosome species by tsetse flies with each of them carrying one trypanosome species. Indeed, tsetse flies carrying mono-specific trypanosome can sequentially infect the same animal through several blood meals [56]. This hypothesis is supported by the fact that cattle can stay in tsetse fly infested areas for several years. Being permanently exposed to tsetse fly bites, the probability to be bitten and to acquire different trypanosome species from tsetse flies carrying mono-specific trypanosome infection is high. Nonetheless, the probability for a bovine to acquire mixed infections during a single blood meal taken by tsetse fly carrying several trypanosome species cannot be ruled out [5, 57]. Our results showing that most mixed infections involved *T. congolense* and trypanosomes of the sub-genus *Trypanozoon* are consistent with data of previous studies highlighting a high proportion of such mixed infections in domestic animal taxa from HAT foci of Côte d'Ivoire and Cameroon [12, 13, 54, 58]. The presence of mixed infections has some implications on the transmission and the epidemiology of AAT. During a single blood meal, an uninfected tsetse fly can acquire different trypanosome species. If these later settle and develop until the metacyclic forms, an infected tsetse fly can therefore transmit these trypanosomes to healthy mammals during single blood meal. However, the bottleneck reported during the establishment and development of trypanosomes in tsetse could select some trypanosome species. If such selection occurs, some trypanosome species will reach the metacyclic forms while others could not. Only those that reach the metacyclic forms will be transmitted to mammals and in consequence, their prevalence will be higher in infected animals compared to other species.

The mixed infections identified in cattle and involving mostly *T. congolense* and trypanosomes of the sub-genus *Trypanozoon* have some impacts on the clinical evolution of the disease. Mixed infections induce interactions between trypanosome species and could have

important evolutionary effects on both hosts and parasites [59]. In trypanosomosis, mixed infections have been reported to be advantageous for host compared to single infections. Indeed, hosts infected with both virulent and non-virulent *T. brucei* strains survived significantly longer than those infected with the more virulent strain alone, despite having received the cumulative infecting dose [59]. It seems that co-infection with a less virulent strain significantly enhances host survival by suppressing the density of the more virulent strain, whose degree of impact ultimately determines the physical condition of the host [59, 60]. Moreover, the competition arising amongst co-infecting trypanosome species could have an impact on the transmission dynamics because it could determine which species could develop to meta-cyclic forms and hence, what species will be further transmitted. Nevertheless, Dwinger *et al.* [61] showed that if the first infections is due to *T. congolense*, there is no effect on the susceptibility of animals to trypanosomes of the subgenus *Trypanozoon*. This could explain the high proportion of trypanosomes of the subgenus *Trypanozoon* in most mixed infections recorded in cattle.

Although little investigations have been undertaken on cattle trypanosomosis in HAT foci of west and central Africa, the identification of different trypanosome species in this animal taxa indicates its epidemiological importance for AAT. This highlights the ability of cattle to maintain and ensure the transmission of different pathogenic trypanosome species in various epidemiological settings. Improving animal health, agriculture production and the economy of smallholder farmers in Chadian HAT foci requires to fight against trypanosome infections in cattle.

In this study, primers specific for *T. b. brucei* and *T. evansi* were not used to identify these trypanosome species. Amongst trypanosome of the subgenus *Trypanozoon*, the main trypanosome species circulating in cattle of the three HAT foci remains therefore unknown as well as the anemic status of cattle. Moreover, as data related to the movement of herds and the breeding systems were not collected during field surveys, the prevalence of trypanosome infections in sedentary cattle and transhumant ones cannot be inferred. Considered as limits for this study, these unidentified factors constitute major drawbacks for a better understanding of AAT transmission and its epidemiology, but also for the designing of the best control strategies.

## Conclusions

This study revealed natural infections of several trypanosome species in cattle of different HAT foci in the south of Chad. It shows that the transmission patterns of these trypanosome species is similar in the different HAT foci. Based on our results, we do not think that cattle cannot be considered as potential reservoir for human-infective trypanosomes in Chad. The identification of several trypanosome species in cattle highlights the need of developing control strategies for AAT with the overarching goal of improving animal health and the economy of smallholder farmers.

## Acknowledgments

We gratefully acknowledge the support of the University of Dschang, the "Alexander von Humboldt Foundation" and breeders of the three sleeping sickness foci of the south of Chad.

## Author Contributions

**Conceptualization:** Joël Vourchakbé, Sartrien Tagueu Kante, Gustave Simo.

**Data curation:** Arnol Auvaker Zebaze Tiofack, Sartrien Tagueu Kante, Padja Abdoul Barka.

**Formal analysis:** Arnol Auvaker Zebaze Tiofack, Sartrien Tagueu Kante, Padja Abdoul Barka.

**Funding acquisition:** Joël Vourchakbé.

**Methodology:** Joël Vourchakbé, Arnol Auvaker Zebaze Tiofack, Padja Abdoul Barka.

**Supervision:** Gustave Simo.

**Writing – original draft:** Joël Vourchakbé, Gustave Simo.

**Writing – review & editing:** Gustave Simo.

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
