## [Decision Letter · Decision Letter 0]

13 Sep 2022

PONE-D-22-22212Prevalence of different trypanosome species in naturally infected cattle of three sleeping sickness foci of the south of ChadPLOS ONE

Dear Dr. Simo,

Thank you for submitting your manuscript to PLOS ONE. After careful consideration, we feel that it has merit but does not fully meet PLOS ONE’s publication criteria as it currently stands. Therefore, we invite you to submit a revised version of the manuscript that addresses the points raised during the review process.

Please attend to all the concerns raised by the reviewers. The study design and how the research was conducted including how sample sizes were arrived at should be well described. More detailed analystical rocedures such binary logistic regression should be employed to ensure that  more meaning is obtained from the results. The discussion should also be revised so that all the issues that have been identified by the reviewers are attended to. Please resubmit the revised manuscript as advised in this letter.

We look forward to receiving your revised manuscript.

Kind regards,

Martin Chtolongo Simuunza, PhD

Academic Editor

PLOS ONE

Journal Requirements:

Note: HTML markup is below. Please do not edit.]

Reviewers' comments:

Reviewer's Responses to Questions

**Comments to the Author**

1. Is the manuscript technically sound, and do the data support the conclusions?

Reviewer #1: Yes

Reviewer #2: Yes

Reviewer #3: Yes

Reviewer #4: Yes

Reviewer #5: Yes

2. Has the statistical analysis been performed appropriately and rigorously? 

Reviewer #1: Yes

Reviewer #2: I Don't Know

Reviewer #3: Yes

Reviewer #4: No

Reviewer #5: No

3. Have the authors made all data underlying the findings in their manuscript fully available?

Reviewer #1: Yes

Reviewer #2: No

Reviewer #3: Yes

Reviewer #4: No

Reviewer #5: Yes

4. Is the manuscript presented in an intelligible fashion and written in standard English?

Reviewer #1: Yes

Reviewer #2: Yes

Reviewer #3: Yes

Reviewer #4: Yes

Reviewer #5: Yes

5. Review Comments to the Author

Reviewer #1: This is a valuable addition to the literature and research effort on animal trypanosomiasis in endemic countries. The MS is well written and presented for the most part. I found the discussion overlong for my taste at 4+ pages, but on the other hand this gave space for thorough discussion and all material was relevant. I have the following suggestions for improvement, many just tweaks to improve readability:

Title: could be more informative "Prevalence of pathogenic trypanosome species in..."

Abstract line 22: cattle is a plural word, use animal instead. Body condition (not plural); ditto line 32. Line 28 no cattle were found...Line 36 expand abbreviation AAT - not previously used?

Intro: Line 69 constitute (livestock is a plural word).

Methods: Line 105, 110, 116 replace peasant with "subsistence". Line 109 replace precipitations with "annual rainfall". Line 123/4 rewrite to clarify "In each village, only cattle that had been exposed to tsetse fly bites by spending at least 3 months in the study zone were included and randomly selected for sampling. From each selected animal,...". Line 128 Delete "Moreover" = unnecessary. Line 141 replace associated to with "supplemented with". Line 143 Clarify whether this procedure was done with all 1466 samples or just the CTC positives. Line 225 P value was <0.05.

Results: Line 230 rewrite "were aged 2 years or over". Line 231/2 Delete "about" = unnecessary; body condition (not plural). Line 307 Moissala spelling. Line 344, 349 "one triple infection", "the last double infection" (not plural).

Table 1, footnote line 250 trypanosomosis.

Table 2 - age groups "<2 and >2 years" leaves out cattle of 2 years exactly - which category includes them?

Discussion: Line 389 delete "too" - not necessary. Line 390 "no cattle were found...". Line 404 "this test is unable to..." - there are certainly ways to distinguish T.evansi from T. brucei, and here you have used a specific PCR test for T.b.gambiense. Line 413 T.equiperdum?? not very likely considering this is an equine tryp transmitted via coitus? or do you envisage some strange transmission scenarios in Chad? Line 420 discrepancies also could be due to susceptibility of different livestock to T.brucei infection? Similarly for T.congolense forest and savannah line 444-449. Line 428 indicate (not plural). Line 432 italics submorsitans. Line 469-470 female and male cattle - no plural needed. Line 483 surely poor body condition can be attributed to trypanosome infection - it's a sign of the disease. Line 501 "for a bovine..." Line 543 smallholder is one word.

General: Might be useful to include a map of Chad with 3 foci locations. Throughout - subgenus does not need a hyphen.

Reviewer #2: The ms entitled « Prevalence of different trypanosome species in naturally infected cattle of three

sleeping sickness foci of the south of Chad » aims to provide information on the prevalence of trypanosomes in randomly selected cattle within sleeping sickness foci in Chad. Although the idea is very good and interesting, and the ms is clearly presented, there are severe lacks that need to be addressed before this can be published. In particular it is very difficult to understand why the authors do not mention and discuss the importance of a vector control campaign in the Mandoul that has been published and the likely impact on their results. Another limit, linked to this, is that it is difficult to draw any conclusion on the epidemiology without knowing the sedentary or transhumant status of the cattle, which have unfortunately not been taken into account in their study design.

Mahamat et al (2017) published a paper entitled « Adding tsetse control to medical activities contributes to decreasing transmission of sleeping sickness in the Mandoul focus (Chad) » where they show a likely disappearance of tsetse flies from 2016 onwards as a consequence of a vector control campaign using tiny targets (https://journals.plos.org/plosntds/article/figure?id=10.1371/journal.pntd.0005792.g003).

Even more, Rock et al., 2022 (Rock et al. Infectious Diseases of Poverty (2022) 11:11

https://doi.org/10.1186/s40249-022-00934-8) , using a modelling approach, concluded that transmission of T. b. gambiense had probably been interrupted since 2015, so well before the sampling described for this present ms (2019). Again, I hardly understand why the authors do not even cite these references, and do not discuss the likely impact of this on their results, at least for the Mandoul g-HAT focus.

As an example, in the Mandoul, given the likely disappearance of tsetse and of Tbg, the expectation is to find zero tsetse transmitted trypanosomes in sedentary animals. The occurrence of trypanosomes may be possible in animals that are transhumant. Unfortunately the authors did not design their study in a way allowing testing and discussing this. Alternatively, it then becomes quite tricky to explain the author’s results describing so many tsetse transmitted trypanosomes in an area where there is no longer any possibility of cyclical transmission, based on these papers.

As an example, how to reconcile, line 306 « The highest prevalence (8.3%) of trypanosomes of the sub-genus Trypanozoon was obtained in cattle from the Mandoul HAT focus » ? Would there be any possibility of false positives in their results ? what does mean a positive PCR, does this mean an active trypanosome infection ? not so sure, by the way…there may be other hypotheses to explain this result (transhumant cattle/Trypanozoon trypanosomes that are not T. b. gambiense/etc.), but ignoring these references is misleading for this study, at least for the situation in the Mandoul.

Reviewer #3: The article which discusses the prevalence of trypanosomosis in the HAT foci of Southern Chad primarily allayed the fears of cattle serving as reservoir of infection to humans. The manuscript was written in simple, coherent and easy to read language. The methods used were quite clearly explained and helps to understand the investigation carried out. i commend the authors for this.

introduction

Line 45: delete..."parasitic diseases" and replace with ..."parasites"

Line 137: replace entire subheading with "Parasitological tests"

Line 223: How did you compare the prevalence of different trypanosome species with Chi-square? Could you be talking about the associations of the variables with positive trypanosome results from the different study areas?

Line 230: replace ..."had" with "were"

Line 249-253: Describe "CT"

Line 259: The percentage age prevalence of positive RDTs was repeated. Delete the repetition.

Line 320: replace "having" with "being"

Line 391-397: What is the author's view on sensitivity and specificity of the RDT in excluding Trypanosoma brucei gambiense?

Line 474-479: I would agree more with the fact that the older cattle are more exposed than the younger cattle a reason for the high prevalence. it might be interesting to note that being immunologically immature connote a higher susceptibility in the younger than the older cattle.

Line 480-488: Many factors may predispose cattle to poor body conditions namely; other concurrent parasitic infections particularly helminth infections in addition to their nutritional status. i had also expected the authors to correlate the body condition of the cattle to the occurrence of anaemia. This is a common feature of trypanosomosis and should have been investigated in line with the body condition scores of these cattle. This could have helped in the discussion of the element of body condition in this study

Reviewer #4: This is a good study. As stated by the authors, identification of trypanosome species and determining their prevalence in animals from different settings are essential for the understanding of AAT epidemiology and the development of control strategies. Much as it explored AAT mainly, it had the potential contribute information on the role of animal reservoirs in the epidemiology of HAT. Current knowledge of T. b. gambiense infections in animals is limited and fragmented.

Main comments/questions

Study design not well described. Was sampling just opportunistic? There is no formula to calculate a representative sample size and therefore we cannot infer if this sample size is reasonable vis -a-vis the cattle population in the region.

The approach to analyse the data is correct but inadequate. Bivariate analyses should be followed by multiple logistic regressions to assess the magnitude of association of the factors with prevalence of infection!

The results for RDT designed for the screening of T.b.gambiense seems interesting. For T. b. gambiense, despite early data generated on its infectivity and transmissibility in animals, the epidemiological significance of any animal reservoir remain at best not well understood. The discussion on this according to me has not been done with sufficient rigour.

The authors opine in line 386-393 that the presence of +ve RDTs but lack of Tbg positive DNAs in cattle could be due to past exposure! Can they cite their sources of this information? How frequent has this phenomenon been observed in cattle? What is the role of cattle as reservoirs of T.b.gambiense overall?

The write up would benefit from edits to make sentences more precise and less wordy. As an example, look at line 372-375. There are three sentences all essentially talking about the same thing! This is rather common throughout the manuscript!

Minor comments

Line 46: Replace “induce” with “cause”

Line 60-61: Should read “the pathogenic animal trypanosomes species…”

Line 64: Use “constrain” instead of “restrain”….

Line 88-90: revise sentence; doesn't read really well!

Line 101: Maro focus be described first, followed by Mandoul and then Moissala

Line 173: check spelling of “congolense”; it’s written as “congolence”. And it occurs in other instances in the manuscript.

Line 426-433: I think G. fuscipes fuscipes in particular has been shown to have a higher transmission index for T. vivax than other species like pallidipes. Moreover T. vivax is known to have the shortest life cycle among all the trypanosome species. So the least predominance of T. vivax seems surprising. Could there be other explanations for this finding in the study area other than vector competence?

Line 450-457: The explanation advanced here can be true for the difference in overall prevalence but may not suffice for the difference in species of trypanosomes observed in the different foci!

Line 474-475: How does “immature immunity of younger cattle” explain lower disease prevalence in younger cattle than older ones? Shouldn’t it be the contrary???

Reviewer #5: Joel et al. submitted an article on "prevalence of different trypanosome species in naturally infected cattle of three sleeping sickness foci of the south of Chad". The authors did a great job which will be an interesting read among the scientific community. However, there are few points that need to be addressed which have been highlighted in the pdf. The major limitation in the work is the failure to speciate the "Trypanozoon" group of trypanosomes identified. This is a major setback in understanding the full epidemiology and best control strategies. Otherwise, the authors have done justice to most part of the work.

Important points

Line 23: Is there any correlation between the HAT foci (location) and AAT infections observed? Or it's there any impact of this location on the AAT species identified?

Line 293: Did the authors differentiate the Trypanozoon group at all? The TBR primers will identify the T. brucei brucei species. The T. evansi primer is also available. The high Trypanozoon group could be dominated by a trypanosome species, which could give us the idea of the predominant tsetse species responsible for transmission in a bid to control the disease.

Line 306: You'd observe that the distribution is even. Knowing the species composition would add to the knowledge if it can be done.

Line 378: The HAT foci considered in Chad, is it a tsetse-free belt area or otherwise? In the literature, you didn't report the distribution of tsetse flies caught in that area before.

Line 440-441: Comparing with equine result from another study doesn't add anything to this explanation. That comparison should be deleted

The 95% confidence interval should be included in all percentages.

The discussion is too long and contains so many unnecessary information.

There's no proper reporting of statistical inferential analysis in the manuscript (%95 CI, P-value)

6. PLOS authors have the option to publish the peer review history of their article (what does this mean?). If published, this will include your full peer review and any attached files.

Reviewer #1: No

Reviewer #2: No

Reviewer #3: No

Reviewer #4: **Yes: **Dr Robert Opiro

Department of Biology

Gulu University, Uganda

Reviewer #5: **Yes: **Dr Paul Odeniran

---

## [Author Response · Author response to Decision Letter 0]

18 Oct 2022

Answer to reviewer comments

Comments to the Author

Reviewer #1

This is a valuable addition to the literature and research effort on animal trypanosomiasis in endemic countries. The MS is well written and presented for the most part. I found the discussion overlong for my taste at 4+ pages, but on the other hand this gave space for thorough discussion and all material was relevant.

1- I have the following suggestions for improvement, many just tweaks to improve readability: Title: could be more informative "Prevalence of pathogenic trypanosome species in..."

Answer

The title has been modified as suggested by the reviewer

2- Abstract line 22: cattle is a plural word, use animal instead.

Answer

The correction was done as suggested by the reviewer

3- Body condition (not plural); ditto line 32. Line 28 no cattle were found...Line 36 expand abbreviation AAT - not previously used?

Answer

The correction was done as suggested by the reviewer

4- Intro: Line 69 constitute (livestock is a plural word).

Answer

The correction was done as suggested by the reviewer

5- Methods: Line 105, 110, 116 replace peasant with "subsistence".

Answer

This was done

6- Line 109 replace precipitations with "annual rainfall".

Answer

The correction was done as suggested

7- Line 123/4 rewrite to clarify "In each village, only cattle that had been exposed to tsetse fly bites by spending at least 3 months in the study zone were included and randomly selected for sampling. From each selected animal,...".

Answer

The correction was done as suggested by the reviewer

8- Line 128 Delete "Moreover" = unnecessary.

Answer

The correction was done as suggested by the reviewer

9- Line 141 replace associated to with "supplemented with".

Answer

This correction was done as suggested by the reviewer

10- Line 143 Clarify whether this procedure was done with all 1466 samples or just the CTC positives.

Answer

The sentence has been rephrased as: “The remaining blood sample was centrifuged at 13,000 rpm for 5 min and the buffy coat was collected from all samples and then, transferred into labelled microtubes.”

11- Line 225 P value was <0.05.

Answer

This correction was done as suggested by the reviewer

12- Results: Line 230 rewrite "were aged 2 years or over".

Answer

This correction was done as suggested by the reviewer

13- Line 231/2 Delete "about" = unnecessary; body condition (not plural).

Answer

This correction was done as suggested by the reviewer

14- Line 307 Moissala spelling.

Answer

The spelling was corrected and “Moissala” has been replaced by “Moïssala”

15- Line 344, 349 "one triple infection", "the last double infection" (not plural).

Answer

These corrections were done as suggested by the reviewer

16- Table 1, footnote line 250 trypanosomosis.

Answer

This correction was done as suggested by the reviewer

17- Table 2 - age groups "<2 and >2 years" leaves out cattle of 2 years exactly - which category includes them?

Answer

The corrections were made: we have “< 2 years” and then “2 years or over”

18- Discussion: Line 389 delete "too" - not necessary.

Answer

This correction was done as suggested

19- Line 390 "no cattle were found...".

Answer

“No cattle” have been replaced by “no animal”

20- Line 404 "this test is unable to..." - there are certainly ways to distinguish T.evansi from T. brucei, and here you have used a specific PCR test for T.b.gambiense.

Answer

This section has been rewritten as:

“Despite the higher specificity of the PCR-based method, the differentiation of trypanosomes of the subgenus Trypanozoon that include T. b. rhodesiense, T. b. brucei, T. b. gambiense, T. evansi and T. equiperdum remains a challenge because no single test is able to unequivocally distinguish these trypanosome species [35].”

21- Line 413 T.equiperdum?? not very likely considering this is an equine tryp transmitted via coitus? or do you envisage some strange transmission scenarios in Chad?

Answer

We thank the reviewer for these remarks. This section has been rephrased and the following sentence has been included: “However, the presence of T. equiperdum is unlikely in cattle because it is an equine trypanosomes transmitted via coitus.”

22- Line 420 discrepancies also could be due to susceptibility of different livestock to T.brucei infection? Similarly for T.congolense forest and savannah line 444-449.

Answer

The point raised by the reviewer has been taken into consideration by including the statement suggested by the reviewer. The sentence has been rephrased as:

- “The discrepancies between these results could be explained not only by the breeding system that varies according to animal taxa and HAT foci, but also by the differences in the susceptibility of animal species to trypanosomes of the subgenus Trypanozoon.”

- “Nevertheless, the differences in susceptibility of different animal taxa to T. congolense infections cannot be ruled out.”

23- Line 428 indicate (not plural).

Answer

The correction was done as indicated by the reviewer

24- Line 432 italics submorsitans.

Answer

The correction was done as suggested

25- Line 469-470 female and male cattle - no plural needed.

Answer

The correction was done as suggested

26- Line 483 surely poor body condition can be attributed to trypanosome infection - it's a sign of the disease.

Answer

We thanks the reviewer for this remark. The following sentence has been added to the section: “In fact, poor body condition can be attributed to trypanosome infections because it is a sign of AAT in livestock.”

27- Line 501 "for a bovine..."

Answer

The correction was done as suggested

28- Line 543 smallholder is one word.

Answer

The correction was done as suggested

- General: Might be useful to include a map of Chad with 3 foci locations. Throughout - subgenus does not need a hyphen.

Answer

For the map of Chad containing the 3 HAT fcoi, the reference Vourchakbe et al. (2020) was added. In this Vourchakbe et al. paper, the map showing these HAT foci has been well highlighted.

Reviewer #2: 

The ms entitled « Prevalence of different trypanosome species in naturally infected cattle of three sleeping sickness foci of the south of Chad » aims to provide information on the prevalence of trypanosomes in randomly selected cattle within sleeping sickness foci in Chad. Although the idea is very good and interesting, and the ms is clearly presented, there are severe lacks that need to be addressed before this can be published. In particular it is very difficult to understand why the authors do not mention and discuss the importance of a vector control campaign in the Mandoul that has been published and the likely impact on their results.

Answer

We agree with the reviewer comments. The point raised has been taken into consuderation and the paragraph bellow has been included in the discussion:

“The highest prevalence of trypanosomes of the subgenus Trypanozoon in cattle from the Mandoul HAT focus is surprising because the “tiny targets” deployed in this focus for vector control during three consecutive years (2014 to 2016) before our sampling were expected to stop trypanosomes’ transmission (Mahamat et al., 2017). This hypothesis was strengthened by results of mechanistic transmission model suggesting that HAT transmission would have been interrupted in 2015 due to intensified interventions in the Mandoul HAT focus (Rock et al., 2022). However, from 2016 to 2020, HAT cases have been reported in all Chadian HAT foci including the Mandoul HAT focus where the number of HAT cases decreased while a slight increase was observed in the Maro HAT focus (Franco et al., 2022). This continuous detection of HAT cases in the Mandoul HAT focus indicates that a slight cyclical transmission of trypanosomes may still occurs despite vector control operations. Nonetheless, the probability of mechanical transmission of trypanosomes of the subgenus Trypanozoon such as T. evansi cannot be excluded. In addition to that, the movement of herders with some infected cattle in the search of pastures during transhumance phenomenon is another factor that could explain the highest prevalence of trypanosomes of the subgenus Trypanozoon in cattle from the Mandoul HAT focus.”

- Another limit, linked to this, is that it is difficult to draw any conclusion on the epidemiology without knowing the sedentary or transhumant status of the cattle, which have unfortunately not been taken into account in their study design.

Answer

We agree with the reviewer comments. It is true that during the study design, we did not include the sedentary or transhumant status of the cattle. It is not part of our objective. However, this point raised by the reviewer was considered as one limit of the study. This limit was reported at the end of the discussion as:

“Moreover, as data related to the movement of herds and the breeding systems were not collected during field surveys, the prevalence of trypanosome infections in sedentary cattle and transhumant ones cannot be inferred. These factors can be considered as limits of this study. These limits constitute therefore a brake for a better understanding of AAT transmission and its epidemiology, but also for the designing of the best control strategies.

- Mahamat et al (2017) published a paper entitled « Adding tsetse control to medical activities contributes to decreasing transmission of sleeping sickness in the Mandoul focus (Chad) » where they show a likely disappearance of tsetse flies from 2016 onwards as a consequence of a vector control campaign using tiny targets (https://journals.plos.org/plosntds/article/figure?id=10.1371/journal.pntd.0005792.g003). Even more, Rock et al., 2022 (Rock et al. Infectious Diseases of Poverty (2022) 11:11 https://doi.org/10.1186/s40249-022-00934-8) , using a modelling approach, concluded that transmission of T. b. gambiense had probably been interrupted since 2015, so well before the sampling described for this present ms (2019). Again, I hardly understand why the authors do not even cite these references, and do not discuss the likely impact of this on their results, at least for the Mandoul g-HAT focus. As an example, in the Mandoul, given the likely disappearance of tsetse and of Tbg, the expectation is to find zero tsetse transmitted trypanosomes in sedentary animals.

- Answer

The references mentioned by the reviewer have been included in the manuscript in the paragraph discussion the impact of vector contol in the Mandoul HAT focus:

“The highest prevalence of trypanosomes of the subgenus Trypanozoon in cattle from the Mandoul HAT focus is surprising because the “tiny targets” deployed in this focus for vector control during three consecutive years (2014 to 2016) before our sampling were expected to stop trypanosomes’ transmission (Mahamat et al., 2017). This hypothesis was strengthened by results of mechanistic transmission model suggesting that HAT transmission would have been interrupted in 2015 due to intensified interventions in the Mandoul HAT focus (Rock et al., 2022). ”

- The occurrence of trypanosomes may be possible in animals that are transhumant. Unfortunately the authors did not design their study in a way allowing testing and discussing this. Alternatively, it then becomes quite tricky to explain the author’s results describing so many tsetse transmitted trypanosomes in an area where there is no longer any possibility of cyclical transmission, based on these papers.

Answer

The occurrence of trypanosomes has been discussed and one paragraph has been decidated to that as:

“The highest prevalence of trypanosomes of the subgenus Trypanozoon in cattle from the Mandoul HAT focus is surprising because the “tiny targets” deployed in this focus for vector control during three consecutive years (2014 to 2016) before our sampling were expected to stop trypanosomes’ transmission (Mahamat et al., 2017). This hypothesis was strengthened by results of mechanistic transmission model suggesting that HAT transmission would have been interrupted in 2015 due to intensified interventions in the Mandoul HAT focus (Rock et al., 2022). However, from 2016 to 2020, HAT cases have been reported in all Chadian HAT foci including the Mandoul HAT focus where the number of HAT cases decreased while a slight increase was observed in the Maro HAT focus (Franco et al., 2022). This continuous detection of HAT cases in the Mandoul HAT focus indicates that a slight cyclical transmission of trypanosomes may still occurs despite vector control operations. Nonetheless, the probability of mechanical transmission of trypanosomes of the subgenus Trypanozoon such as T. evansi cannot be excluded. In addition to that, the movement of herders with some infected cattle in the search of pastures during transhumance phenomenon is another factor that could explain the highest prevalence of trypanosomes of the subgenus Trypanozoon in cattle from the Mandoul HAT focus.”

As an example, how to reconcile, line 306 « The highest prevalence (8.3%) of trypanosomes of the sub-genus Trypanozoon was obtained in cattle from the Mandoul HAT focus » ? Would there be any possibility of false positives in their results ? what does mean a positive PCR, does this mean an active trypanosome infection ? not so sure, by the way…there may be other hypotheses to explain this result (transhumant cattle/Trypanozoon trypanosomes that are not T. b. gambiense/etc.), but ignoring these references is misleading for this study, at least for the situation in the Mandoul.

Answer

Results reporting “The highest prevalence (8.3%) of trypanosomes of the sub-genus Trypanozoon was obtained in cattle from the Mandoul HAT focus” have been discussed as indicated in the paragraph mentioned above

Reviewer #3:

The article which discusses the prevalence of trypanosomosis in the HAT foci of Southern Chad primarily allayed the fears of cattle serving as reservoir of infection to humans. The manuscript was written in simple, coherent and easy to read language. The methods used were quite clearly explained and helps to understand the investigation carried out. i commend the authors for this.

Introduction

1- Line 45: delete..."parasitic diseases" and replace with ..."parasites"

Answer

We don’t agree with the reviewer suggestion because “parasitic diseases” refer to African trypanosomoses which are diseases. In this sentence, it was written “African trypanosomoses” instead of “African trypanosomes” as indicated by the reviewer.

2- Line 137: replace entire subheading with "Parasitological tests"

Answer

The replacement was done as suggested

3- Line 223: How did you compare the prevalence of different trypanosome species with Chi-square? 

Answer

The test used here was the k-proportion test or chi-square test of equality of proportion which determine if k proportions can turn out to be all equal (null hypothesis H0) or if at least two proportions are different (alternative hypothesis Ha). To clarify this statement, the sentence was modified as follow: “The k-proportion test or chi-square test of equality of proportion was used to compare the prevalence of different trypanosome species and also the prevalence of trypanosome infections between HAT foci.”

Could you be talking about the associations of the variables with positive trypanosome results from the different study areas?

Answer

We are not talking about the association. Our interest was to compare the infection rates using the K-proportions test or the chi-square test of equality of proportion according to the HAT foci.

4- Line 230: replace ..."had" with "were"

Answer

The sentence has been rephrased as: “Of these 1466 cattle, 1127 (76.9%) were aged two years or over while 339 (23.1%) were calves (Table 2)”. This modification was suggested by another reviewer

5- Line 249-253: Describe "CT"

Answer

This was done by inserting in the legend the following statement: TC: Trypanosoma congolense (including Trypanosoma congolense savannah and forest);

6- Line 259: The percentage age prevalence of positive RDTs was repeated. Delete the repetition.

Answer

The sentence has been rephrased as: “ The percentage of positive RDT was 3.1% in males as well as in females, 2.06% in calves and 3.37% in older cattle”.

8- Line 320: replace "having" with "being"

Answer

This replacement was done as suggested

9- Line 391-397: What is the author's view on sensitivity and specificity of the RDT in excluding Trypanosoma brucei gambiense?

Answer

The author’s view on the sensitivity and specificity of the RDT has been given in the sentence: “Our results highlight the low sensitivity and specificity of RDT for the detection of T. b. gambiense infections in cattle.”

10- Line 474-479: I would agree more with the fact that the older cattle are more exposed than the younger cattle a reason for the high prevalence. it might be interesting to note that being immunologically immature connote a higher susceptibility in the younger than the older cattle.

Answer

We thank the reviewer for the remark made. Statement regarding the immunity of young cattle we deleted.

11- Line 480-488: Many factors may predispose cattle to poor body conditions namely; other concurrent parasitic infections particularly helminth infections in addition to their nutritional status. i had also expected the authors to correlate the body condition of the cattle to the occurrence of anaemia. This is a common feature of trypanosomosis and should have been investigated in line with the body condition scores of these cattle. This could have helped in the discussion of the element of body condition in this study.

Answer

This section has been improved with the rephrase and addition of the following sentences:

- “Nevertheless, it is important to point out that animals can become emaciated due to poor nutritional status or other concurrent parasitic infections like helminth infections.”

- “Although the occurrence of anaemia is a common feature of AAT, the anemic status of cattle was not recorded in the present study. This limitation did not allow to assess the correlation between the body condition and anemic status of cattle.”

Reviewer #4:

This is a good study. As stated by the authors, identification of trypanosome species and determining their prevalence in animals from different settings are essential for the understanding of AAT epidemiology and the development of control strategies. Much as it explored AAT mainly, it had the potential contribute information on the role of animal reservoirs in the epidemiology of HAT. Current knowledge of T. b. gambiense infections in animals is limited and fragmented.

- Main comments/questions

Study design not well described. Was sampling just opportunistic? There is no formula to calculate a representative sample size and therefore we cannot infer if this sample size is reasonable vis -a-vis the cattle population in the region.

Answer

A section has been included for the sampling size estimation as:

“Sample size estimation

 For this study, a stratified sampling strategy was used to select herds and individual animal per herd. Only herds with a minimum of 10 cattle were included. Cattle were sampled by herd and in each herd, blood samples were collected in about 20% of animals. However, more than 20% of animals of some herds were sampled due to the interests and cooperation of some herders and advice from veterinarians. From each chosen herd, the selection of cattle to sample was performed as described by Asgedom et al. (2016) using a systematic random sampling technique. The sample size was estimated as described by Thursfield (2007).”

- The approach to analyse the data is correct but inadequate. Bivariate analyses should be followed by multiple logistic regressions to assess the magnitude of association of the factors with prevalence of infection!

Answer

Thanks for the point raised by the reviewer. Section dedicated to data analysis has been improved by performing a multiple logistic regression test to assess the association between the different factors and trypanosome infections. The sentence below was added to this section: 

“Multiple logistic regression models were used to estimate odd ratio (OR) and 95% confidence intervals (CI) for the association between HAT foci, body condition, age groups, sex and trypanosome infections.”

- In the result section, values of odd ration were added in tables (Tables 2) and the test. Some section of the discussion have been reviewed as: “These results are strengthened by the low values of odd ratio indicating decrease risk to be carriers of trypanosome infections for cattle having good body condition compared to those with medium and poor body condition.”

- The results for RDT designed for the screening of T.b.gambiense seems interesting. For T. b. gambiense, despite early data generated on its infectivity and transmissibility in animals, the epidemiological significance of any animal reservoir remain at best not well understood. The discussion on this according to me has not been done with sufficient rigour.

The authors opine in line 386-393 that the presence of +ve RDTs but lack of Tbg positive DNAs in cattle could be due to past exposure! Can they cite their sources of this information? How frequent has this phenomenon been observed in cattle?

Answer

Section of the discussion dedicated to results of RDT and T. b. gambiense infections has been rewritten. Some sentences have been rephrased and new sentences have been included as indicated in the following sentences: “Our RDT results suggest that 3.1% of analyzed cattle have probably been infected by T. b. gambiense. This is low compared to 18.9% and 19.4% recorded respectively in equines and small ruminants of the same HAT foci [15, 16]. The positivity of RDT could be explained by the fact that cattle can sustain T. b. gambiense infections for more than 50 days (Joshua et al., 1983; Moloo et al., 1986). Moreover, the possibility for cattle to be infected by T. b. gambiense have been demonstrated by cyclical development of T. b. gambiense from cattle (Moloo et al., 1986) as well as the isolation from cattle, in Nigeria, of trypanosomes of the subgenus Trypanozoon that were resistant to human serum (Joshua et al., 1983). Despite the positivity of RDT, no animal was infected by T. b. gambiense since all PCRs targeting this parasite were negative. These results could be explained by the fact that a positive PCR can be interpreted as an active infection although problems of reproductibility of PCR for the diagnostic of HAT have been highlighted by Kofﬁ et al [36]. For cattle that were positive to RDT and negative for the PCR targeting T. b. gambiense, the possibility of past infections and self-cure cannot be ruled out because such phenomena have been observed in other animal taxa like pigs (Penchenier et al., 2005).”

- What is the role of cattle as reservoirs of T.b.gambiense overall?

Answer

Results of this study indicate that cattle cannot be considered as potential reservoir for T. b. Gambiense. This was indicated in follwing sentence of the discussion: “Cattle cannot therefore be considered as potential animal reservoir for the gambiense-HAT in the south of Chad.”

- The write up would benefit from edits to make sentences more precise and less wordy. As an example, look at line 372-375. There are three sentences all essentially talking about the same thing! This is rather common throughout the manuscript!

Answer

Sentences of lines 372-375 have been combined in as: “From the 96 cattle carrying trypanosomes of the sub-genus Trypanozoon, the two sets of primers used to identify T. b. gambiense infections did not amplify DNA fragment of this trypanosome sub-species: no cattle of the three HAT foci was therefore found with T. b. gambiense infections”.

Minor comments

- Line 46: Replace “induce” with “cause”

Answer

This replacement was done as suggested

- Line 60-61: Should read “the pathogenic animal trypanosomes species…”

Answer

The correction was done as suggested

- Line 64: Use “constrain” instead of “restrain”….

Answer

The correction was done as suggested

- Line 88-90: revise sentence; doesn't read really well!

Answer

This section has been rewritten as: “In the progressive control pathway that was recently defined for AAT, it is advocated to enhance research aiming to understand the risks of trypanosomosis. Appearing as the first step for the process that will lead to effective control of AAT, a better understanding these risks is essential to guide the selection of priority intervention areas [17].”

- Line 101: Maro focus be described first, followed by Mandoul and then Moissala

Answer

The order of the description was modified as suggested by this reviewer

- Line 173: check spelling of “congolense”; it’s written as “congolence”. And it occurs in other instances in the manuscript.

Answer

The corrections were done 

- Line 426-433: I think G. fuscipes fuscipes in particular has been shown to have a higher transmission index for T. vivax than other species like pallidipes. Moreover T. vivax is known to have the shortest life cycle among all the trypanosome species. So the least predominance of T. vivax seems surprising. Could there be other explanations for this finding in the study area other than vector competence?

Answer 

The most likely explanation could be the fact that very few mammals as well as tsetse flies are infected by T. vivax. With few infected tsetse or mammals with and abscence of an important reservoir of T. vivax, the transmission of this parasite is probably low.

- Line 450-457: The explanation advanced here can be true for the difference in overall prevalence but may not suffice for the difference in species of trypanosomes observed in the different foci!

Answer

We agree with point raised by the reviewer. This section has been improved with the addition of the following sentences:

- As already mentioned above, some differences in the susceptibility of trypanosomes to animal taxa are additional arguments to explain the variations observed in the prevalence of trypanosome infections.

- Moreover, the high pathogenicity of some trypanosome strains for some animal taxa cannot be excluded. 

- Line 474-475: How does “immature immunity of younger cattle” explain lower disease prevalence in younger cattle than older ones? Shouldn’t it be the contrary???

Answer

We agree with the reviewer comment. The statement concerning immature immunity of younger cettle was deleted. The sentence has been rephrased as: “These results could be explained by the higher exposure of older cattle to tsetse bites compared to younger ones [42]. In fact, younger cattle are often kept in the farmstead and do not venture far for grazing and watering.”

Reviewer #5:

Joel et al. submitted an article on "prevalence of different trypanosome species in naturally infected cattle of three sleeping sickness foci of the south of Chad". The authors did a great job which will be an interesting read among the scientific community. However, there are few points that need to be addressed which have been highlighted in the pdf. The major limitation in the work is the failure to speciate the "Trypanozoon" group of trypanosomes identified. This is a major setback in understanding the full epidemiology and best control strategies. Otherwise, the authors have done justice to most part of the work.

Important points

- Line 23: Is there any correlation between the HAT foci (location) and AAT infections observed? Or it's there any impact of this location on the AAT species identified?

Answer

Looking at the distribution of trypanosomes species according to HAT, the patterns of infections very according to trypanosome species and HAT foci. For instance, no significant difference between HAT foci was observed for trypanosomes of the subgenus Trypanozoon. This point has been highlighted and discussed respectively in the result and the discussion.

- “The highest prevalence (8.3%) of trypanosomes of the sub-genus Trypanozoon was obtained in cattle from the Mandoul HAT focus followed by those from Moïssala (6.6%) and Maro (4.9%) HAT foci. Between HAT foci, no significant difference (χ2 = 4.79; p = 0.091) was observed in the prevalence of trypanosomes of the sub-genus Trypanozoon”

- “Our results highlighting no significant difference in the prevalence of trypanosomes of the subgenus Trypanozoon suggests similar transmission patterns in the three Chadian HAT foci.”

 For other trypanosome species, the patterns of infections were different as described in the result section entitled: “Prevalence of trypanosome species according to HAT foci”

- Line 293: Did the authors differentiate the Trypanozoon group at all? The TBR primers will identify the T. brucei brucei species. The T. evansi primer is also available. The high Trypanozoon group could be dominated by a trypanosome species, which could give us the idea of the predominant tsetse species responsible for transmission in a bid to control the disease.

Answer

We agree with the reviewer that it is impossible at this stage to identify the dominated trypanosome species. Although specific primers exist for T. evansi, the TBR primers designed by Moser et al. (1989) is not specific to T. b. brucei. These primers amplify DNA of all trypanosomes of the subgenus Trypanozoon. The point raised by the reviewer concerning the undifferentiation of trypanosomes of the subgenus Trypanozoon was consiered as one limit of this study. One section highlighting this limit has been included in the discussion as: “In this study, primers specific for T. b. brucei and T. evansi were not used to identify these trypanosome species. Among trypanosome of the subgenus Trypanozoon, the main trypanosome species that circulates in cattle of these HAT foci remains therefore unknown. This appears as one major limit for the understanding of the transmission and the epidemiology of AAT, but also for the designing of the best control strategies.”

- Line 306: You'd observe that the distribution is even. Knowing the species composition would add to the knowledge if it can be done.

Answer

As already mentioned above, the unspeciation of trypanosomes of the subgenus Trypanozoon was consiered as one limit of the study. Nonetheless, data regarding the distribution and the prevalence of trypanosomes of the subgenus Trypanozoon were discussed as: “Our results highlighting no significant difference in the prevalence of trypanosomes of the subgenus Trypanozoon suggests similar transmission patterns in the three Chadian HAT foci.”

- Line 378: The HAT foci considered in Chad, is it a tsetse-free belt area or otherwise? In the literature, you didn't report the distribution of tsetse flies caught in that area before.

Answer

The HAT foci of Chad are located in tsetse infested areas. One section of the introduction was modified with the following sentences:

- “However, in the Chadian HAT foci that are located in the southern part of the country, environmental conditions are favorable for cattle breeding and tsetse development.”

- “Entomological studies reported several tsetse species including Glossina tachinoides, G. fuscipes fuscipes and G. morsitans submorsitans in HAT foci of Chad.”

-Line 440-441: Comparing with equine result from another study doesn't add anything to this explanation. That comparison should be deleted

Answer

Comparison with equine results was deleted and this section has been rewritten as: “Our results showing a comparable prevalence of T. congolense savannah (2.9%) and T. congolense forest (2.5%) is difficult to explain because the localization of the three Chadian HAT foci in the forest galleries would have suggested higher prevalence of T. congolense forest compared to T. congolense savannah.”

- The 95% confidence interval should be included in all percentages.

Answer

The suggestion has been taken into account and 95%CI have been included in different tables as well as in the text.

- The discussion is too long and contains so many unnecessary information.

Answer

Some section of the discussion have been deleted

- There's no proper reporting of statistical inferential analysis in the manuscript (%95 CI, P-value)

Answer

The 95% CI as well as the P values have been inserted in the tables and the text

---

## [Decision Letter · Decision Letter 1]

1 Nov 2022

PONE-D-22-22212R1Prevalence of pathogenic trypanosome species in naturally infected cattle of three sleeping sickness foci of the south of ChadPLOS ONE

Dear Dr. Simo,

Thank you for submitting your manuscript to PLOS ONE. After careful consideration, we feel that it has merit but does not fully meet PLOS ONE’s publication criteria as it currently stands. Therefore, we invite you to submit a revised version of the manuscript that addresses the points raised during the review process.

One of the reviewers recommends that you make minor revisions to the manuscript as outlined in their attached coments. Please attend to all their comments and return the revised manuscript as adivised in this letter. 

We look forward to receiving your revised manuscript.

Kind regards,

Martin Chtolongo Simuunza, PhD

Academic Editor

PLOS ONE

Journal Requirements:

Reviewers' comments:

Reviewer's Responses to Questions

**Comments to the Author**

1. If the authors have adequately addressed your comments raised in a previous round of review and you feel that this manuscript is now acceptable for publication, you may indicate that here to bypass the “Comments to the Author” section, enter your conflict of interest statement in the “Confidential to Editor” section, and submit your "Accept" recommendation.

Reviewer #1: All comments have been addressed

Reviewer #2: (No Response)

Reviewer #4: All comments have been addressed

Reviewer #5: All comments have been addressed

2. Is the manuscript technically sound, and do the data support the conclusions?

Reviewer #1: (No Response)

Reviewer #2: Partly

Reviewer #4: Yes

Reviewer #5: Yes

3. Has the statistical analysis been performed appropriately and rigorously? 

Reviewer #1: (No Response)

Reviewer #2: Yes

Reviewer #4: Yes

Reviewer #5: Yes

4. Have the authors made all data underlying the findings in their manuscript fully available?

Reviewer #1: (No Response)

Reviewer #2: Yes

Reviewer #4: Yes

Reviewer #5: Yes

5. Is the manuscript presented in an intelligible fashion and written in standard English?

Reviewer #1: (No Response)

Reviewer #2: Yes

Reviewer #4: Yes

Reviewer #5: Yes

6. Review Comments to the Author

Reviewer #1: The authors have addressed all points. The altered spelling of Moissala needs to be consistently applied throughout, not only in Results but also Abstract and Methods.

Reviewer #2: The authors have correctly answered most of the comments made. A couple of points still need clarification/better formulation. In the discusssion, the authors should spend more time discussing the limits of their results and the tools they use, whereas the discussion is very long on sometimes very speculative results. E.g. are they sure that a PCR positive means an active infection ? I am not. Can’t their be any false positive or false negative results ? the authors use RDTs for gambiense in cattle, but what is the performance on this test designed for humans in animals ? do you have any idea ? any reference ? I think these are exemples of points that need to be discussed before interpreting what they may mean epidemiologically…

In their answer, the authors state

« However, from 2016 to 2020, HAT cases have been reported in all Chadian HAT foci including the Mandoul HAT focus where the number of HAT cases decreased while a slight increase was observed in the Maro HAT focus (Franco et al., 2022). This continuous detection of HAT cases in the Mandoul HAT focus indicates that a slight cyclical transmission of trypanosomes may still occurs despite vector control operations. ».

May I clarify here : if Rock et al., 2022 are correct in saying that g-HAT transmission has been interrupted in the Mandoul since 2015, this does not preclude from still finding HAT cases, that would have been infected before the interruption of transmission (g-HAT is a chronic, long lasting disease), or that would come from other locations where there is g-HAT…but it does not indicate that cyclical transmission occurs, because this would be in total contradiction with the beginning of the sentence. So please just remove this sentence otherwise you also to need to discuss this point into details, the rest is ok.

Hence this paragraph should now be (l.446 of the revised version) :

« The highest prevalence of trypanosomes of the subgenus Trypanozoon in cattle from the Mandoul HAT focus is surprising because the “tiny targets” deployed in this focus for vector control during three consecutive years (2014 to 2016) before our sampling were expected to stop trypanosomes transmission (Mahamat et al., 2017). This hypothesis was strengthened by results of mechanistic transmission model suggesting that HAT transmission would have been interrupted in 2015 due to intensified interventions in the Mandoul HAT focus (Rock et al., 2022). However, from 2016 to 2020, HAT cases have been reported in all Chadian HAT foci including the Mandoul HAT focus where the number of HAT cases decreased while a slight increase was observed in the Maro HAT focus (Franco et al., 2022). This continuous detection of HAT cases in the Mandoul HAT focus indicates that a slight cyclical transmission of trypanosomes may still occurs despite vector control operations. Nonetheless, the probability of mechanical transmission of trypanosomes of the subgenus Trypanozoon such as T. evansi cannot be excluded. In addition to that, the movement of herders with some infected cattle in the search of pastures during transhumance phenomenon is another factor that could explain the highest prevalence of trypanosomes of the subgenus Trypanozoon in cattle from the Mandoul HAT focus. »

In additon, the fact that there has been a vector control campaign in the Mandoul must be clarified very early in the ms for the reader. This may be stated in M&M section, line 115 for instance.

Please avoid this kind of arbitrary sentence, especially in your conclusion « No infection of T. b. gambiense was recorded in cattle despite the positivity of RDT used to screen HAT. Cattle cannot be considered as potential reservoir for human-infective trypanosomes in Chad. » please avoid this ! say something like « based on our results, we do not think , bla bla… »

You also spend a very long time discussion your results, but very little time discussing the limits of your results and tools used.

Reviewer #4: The authors have have tried and addressed all my concerns in the initial review! I therefore have no further comments to make except to offer my congratulations to the authors on a job well done

Reviewer #5: The authors have addressed the queries, and provided scientific limitation to their study. The statistical analysis have been performed thoroughly after being raised as a concern in the initial version. This new version is a better one and can be published.

7. PLOS authors have the option to publish the peer review history of their article (what does this mean?). If published, this will include your full peer review and any attached files.

Reviewer #1: No

Reviewer #2: No

Reviewer #4: **Yes: **Robert Opiro

Department of Biology

Gulu University

Reviewer #5: **Yes: **Dr Paul Odeniran

---

## [Author Response · Author response to Decision Letter 1]

3 Nov 2022

Answer to reviewer comments

Reviewer #2: The authors have correctly answered most of the comments made. A couple of points still need clarification/better formulation. In the discusssion, the authors should spend more time discussing the limits of their results and the tools they use, whereas the discussion is very long on sometimes very speculative results. 

- E.g. are they sure that a PCR positive means an active infection ? I am not. Can’t their be any false positive or false negative results ? 

Answer

The point raised by the reviewer as been taken into consideration. The following sentence has been included in the discussion: “It is important to point out that PCR can detect transient infections and a PCR positive result indicates the presence of the corresponding parasite DNA and not necessary an active infection (Herder et al., 2002).”

- The authors use RDTs for gambiense in cattle, but what is the performance on this test designed for humans in animals ? do you have any idea ? any reference ? I think these are exemples of points that need to be discussed before interpreting what they may mean epidemiologically…

Answer

The RDT used in the present study has been already tested on cattle by Matovu et al. (2017). These authors reported also low sensitivity and specificity as obtained in the present study. They reported that these serological tests detect cross-reacting antibodies in cattle. One section of the discussion has been rephrased as: “Results of the present study confirm the low sensitivity and specificity of RDT for the detection of T. b. gambiense infections in cattle as already reported by Matocu et al. [23]. These authors reported that RDTs detect cross-reacting antibodies in cattle.”

- In their answer, the authors state. « However, from 2016 to 2020, HAT cases have been reported in all Chadian HAT foci including the Mandoul HAT focus where the number of HAT cases decreased while a slight increase was observed in the Maro HAT focus (Franco et al., 2022). This continuous detection of HAT cases in the Mandoul HAT focus indicates that a slight cyclical transmission of trypanosomes may still occurs despite vector control operations. ».

May I clarify here : if Rock et al., 2022 are correct in saying that g-HAT transmission has been interrupted in the Mandoul since 2015, this does not preclude from still finding HAT cases, that would have been infected before the interruption of transmission (g-HAT is a chronic, long lasting disease), or that would come from other locations where there is g-HAT…but it does not indicate that cyclical transmission occurs, because this would be in total contradiction with the beginning of the sentence. So please just remove this sentence otherwise you also to need to discuss this point into details, the rest is ok.

Hence this paragraph should now be (l.446 of the revised version) :

« The highest prevalence of trypanosomes of the subgenus Trypanozoon in cattle from the Mandoul HAT focus is surprising because the “tiny targets” deployed in this focus for vector control during three consecutive years (2014 to 2016) before our sampling were expected to stop trypanosomes transmission (Mahamat et al., 2017). This hypothesis was strengthened by results of mechanistic transmission model suggesting that HAT transmission would have been interrupted in 2015 due to intensified interventions in the Mandoul HAT focus (Rock et al., 2022). However, from 2016 to 2020, HAT cases have been reported in all Chadian HAT foci including the Mandoul HAT focus where the number of HAT cases decreased while a slight increase was observed in the Maro HAT focus (Franco et al., 2022). This continuous detection of HAT cases in the Mandoul HAT focus indicates that a slight cyclical transmission of trypanosomes may still occurs despite vector control operations. Nonetheless, the probability of mechanical transmission of trypanosomes of the subgenus Trypanozoon such as T. evansi cannot be excluded. In addition to that, the movement of herders with some infected cattle in the search of pastures during transhumance phenomenon is another factor that could explain the highest prevalence of trypanosomes of the subgenus Trypanozoon in cattle from the Mandoul HAT focus. »

Answer 

This section of the discussion has been rephrased as suggested by the reviewer.

- In additon, the fact that there has been a vector control campaign in the Mandoul must be clarified very early in the ms for the reader. This may be stated in M&M section, line 115 for instance.

Answer 

The following sentence has been added in the Material and method section: “This focus has been subjected to vector control campaign through the deployment of “tiny targets” for vector control during three consecutive years (2014 to 2016) (Mahamat et al., 2017).”

Please avoid this kind of arbitrary sentence, especially in your conclusion « No infection of T. b. gambiense was recorded in cattle despite the positivity of RDT used to screen HAT. Cattle cannot be considered as potential reservoir for human-infective trypanosomes in Chad. » please avoid this ! say something like « based on our results, we do not think , bla bla… »

Answer

The sentence as been rephrased as: “Based on our results, we do not think that cattle cannot be considered as potential reservoir for human-infective trypanosomes in Chad.”

You also spend a very long time discussion your results, but very little time discussing the limits of your results and tools used.

---

## [Decision Letter · Decision Letter 2]

8 Dec 2022

PONE-D-22-22212R2Prevalence of pathogenic trypanosome species in naturally infected cattle of three sleeping sickness foci of the south of ChadPLOS ONE

Dear Dr. Simo,

Thank you for submitting your manuscript to PLOS ONE. After careful consideration, we feel that it has merit but does not fully meet PLOS ONE’s publication criteria as it currently stands. Therefore, we invite you to submit a revised version of the manuscript that addresses the points raised during the review process.

We look forward to receiving your revised manuscript.

Kind regards,

Martin Chtolongo Simuunza, PhD

Academic Editor

PLOS ONE

Journal Requirements:

Additional Editor Comments:

One of the reviewers is still of the view that you have not adequately attended to their comments. I am therefore requesting you to either incoperate their concerns in the manuscript or give a reason why that should not be the case. Then you can resubmit the revised manuscript as advised in this letter.

Reviewers' comments:

Reviewer's Responses to Questions

**Comments to the Author**

1. If the authors have adequately addressed your comments raised in a previous round of review and you feel that this manuscript is now acceptable for publication, you may indicate that here to bypass the “Comments to the Author” section, enter your conflict of interest statement in the “Confidential to Editor” section, and submit your "Accept" recommendation.

Reviewer #1: All comments have been addressed

Reviewer #2: (No Response)

2. Is the manuscript technically sound, and do the data support the conclusions?

Reviewer #1: (No Response)

Reviewer #2: (No Response)

3. Has the statistical analysis been performed appropriately and rigorously? 

Reviewer #1: (No Response)

Reviewer #2: (No Response)

4. Have the authors made all data underlying the findings in their manuscript fully available?

Reviewer #1: (No Response)

Reviewer #2: (No Response)

5. Is the manuscript presented in an intelligible fashion and written in standard English?

Reviewer #1: (No Response)

Reviewer #2: (No Response)

6. Review Comments to the Author

Reviewer #1: (No Response)

Reviewer #2: everything ok now thank you, only one of my earlier comment has not been taken into account, not because of the authors, but because of a typo in earlier comments. Please this paragraph starting l 448 should appear like that :

« The highest prevalence of trypanosomes of the subgenus Trypanozoon in cattle from

the Mandoul HAT focus is surprising because the “tiny targets” deployed in this focus

for vector control during three consecutive years (2014 to 2016) before our sampling

were expected to stop trypanosomes transmission (Mahamat et al., 2017). This

hypothesis was strengthened by results of mechanistic transmission model suggesting

that HAT transmission would have been interrupted in 2015 due to intensified

interventions in the Mandoul HAT focus (Rock et al., 2022). However, from 2016 to

2020, HAT cases have been reported in all Chadian HAT foci including the Mandoul

HAT focus where the number of HAT cases decreased while a slight increase was

observed in the Maro HAT focus (Franco et al., 2022). This continuous detection of

HAT cases in the Mandoul HAT focus indicates that a probability of mechanical transmission of trypanosomes of the subgenus Trypanozoon such as T. evansi cannot be excluded. In addition to that, the movement of herders

with some infected cattle in the search of pastures during transhumance phenomenon

is another factor that could explain the highest prevalence of trypanosomes of the

subgenus Trypanozoon in cattle from the Mandoul HAT focus.

7. PLOS authors have the option to publish the peer review history of their article (what does this mean?). If published, this will include your full peer review and any attached files.

Reviewer #1: No

Reviewer #2: No

---

## [Author Response · Author response to Decision Letter 2]

13 Dec 2022

Answer to reviewer comment

Reviewer #2: everything ok now thank you, only one of my earlier comment has not been taken into account, not because of the authors, but because of a typo in earlier comments. 

Please this paragraph starting l 448 should appear like that :

« The highest prevalence of trypanosomes of the subgenus Trypanozoon in cattle from the Mandoul HAT focus is surprising because the “tiny targets” deployed in this focus for vector control during three consecutive years (2014 to 2016) before our sampling were expected to stop trypanosomes transmission (Mahamat et al., 2017). This hypothesis was strengthened by results of mechanistic transmission model suggesting that HAT transmission would have been interrupted in 2015 due to intensified interventions in the Mandoul HAT focus (Rock et al., 2022). However, from 2016 to 2020, HAT cases have been reported in all Chadian HAT foci including the Mandoul HAT focus where the number of HAT cases decreased while a slight increase was observed in the Maro HAT focus (Franco et al., 2022). This continuous detection of HAT cases in the Mandoul HAT focus indicates that a probability of mechanical transmission of trypanosomes of the subgenus Trypanozoon such as T. evansi cannot be excluded. In addition to that, the movement of herders with some infected cattle in the search of pastures during transhumance phenomenon is another factor that could explain the highest prevalence of trypanosomes of the subgenus Trypanozoon in cattle from the Mandoul HAT focus.

Answer

Although we agree with most sentences of the reviewer suggestions, the sentence before the last has been modified. In this sentence, we don’t agree with the reviewer suggestions because this sentence gives the impression that the continueous detection of HAT cases in the Mandoul HAT focus could be due to mechanical transmission. The sentence suggested by the reviewer was: “This continuous detection of HAT cases in the Mandoul HAT focus indicates that a probability of mechanical transmission of trypanosomes of the subgenus Trypanozoon such as T. evansi cannot be excluded.” It highlights the fact that Trypanosoma brucei gambiense can be mechanically transmitted to human. This has not be demonstrated and the probability of having such transmission to human is unlikely because T. b. gambiense is characterized by a low parasite load in mammals. To clarifiy this point, we have two hypotheses:

1- The HAT cases continueously detected could have been infected several years ago (chronicity of the gambiense-HAT);

2- residual tsetse fly populations could persist after vector control and could ensure biological transmission of trypanosomes to mammals. 

Despite the success of vector control, it is likely that residual tsetse populations can persist in some biotopes and therefore, the possibility of a slight biological transmission of the trypanosomes to human cannot be ruled out. It is for these reasons that the sentences have been rephrased as: “This continuous detection of HAT cases in the Mandoul HAT focus could be explained by the chronicity of the gambiense-HAT since some infected individuals can carry the parasite for many years without being detected. Nevertheless, we cannot rule out the probability of having not only a slight cyclical transmission of trypanosomes to human and mammals by potential residual tsetse fly populations, but also a mechanical transmission of trypanosomes of the subgenus Trypanozoon like T. evansi to cattle [6,7].”

---

## [Editor Report · Decision Letter 3]

14 Dec 2022

Prevalence of pathogenic trypanosome species in naturally infected cattle of three sleeping sickness foci of the south of Chad

PONE-D-22-22212R3

Dear Dr. Simo,

We’re pleased to inform you that your manuscript has been judged scientifically suitable for publication and will be formally accepted for publication once it meets all outstanding technical requirements.

Kind regards,

Martin Chtolongo Simuunza, PhD

Academic Editor

PLOS ONE
---

## [Editor Report · Acceptance letter]

19 Dec 2022

PONE-D-22-22212R3 

Prevalence of pathogenic trypanosome species in naturally infected cattle of three sleeping sickness foci of the south of Chad 

Dear Dr. Simo:

I'm pleased to inform you that your manuscript has been deemed suitable for publication in PLOS ONE. Congratulations! Your manuscript is now with our production department. 

Kind regards, 

on behalf of

Dr. Martin Chtolongo Simuunza 

Academic Editor

PLOS ONE